health and disease and epidemiology

incentives, data sharing, reproducibility, policy, open science, randomized controlled trial

**Author for correspondence:**
Anisa Rowhani-Farid
e-mail: arowhani@rx.umaryland.edu

# Did awarding badges increase data sharing in *BMJ Open*? A randomized controlled trial

Anisa Rowhani-Farid[1,2], Adrian Aldcroft[3]
and Adrian G. Barnett[2]

[1]Department of Pharmaceutical Health Services Research, University of Maryland, Baltimore, MD, USA
[2]School of Public Health and Social Work, Queensland University of Technology, Brisbane, Australia
[3]BMJ Publishing Group, London, UK

AR-F, 0000-0003-3637-2423; AA, 0000-0003-0106-720X;
AGB, 0000-0001-6339-0374

Sharing data and code are important components of reproducible research. Data sharing in research is widely discussed in the literature; however, there are no well-established evidence-based incentives that reward data sharing, nor randomized studies that demonstrate the effectiveness of data sharing policies at increasing data sharing. A simple incentive, such as an Open Data Badge, might provide the change needed to increase data sharing in health and medical research. This study was a parallel group randomized controlled trial (protocol registration: doi:10.17605/OSF.IO/PXWZQ) with two groups, control and intervention, with 80 research articles published in *BMJ Open* per group, with a total of 160 research articles. The intervention group received an email offer for an Open Data Badge if they shared their data along with their final publication and the control group received an email with no offer of a badge if they shared their data with their final publication. The primary outcome was the data sharing rate. Badges did not noticeably motivate researchers who published in *BMJ Open* to share their data; the odds of awarding badges were nearly equal in the intervention and control groups (odds ratio = 0.9, 95% CI [0.1, 9.0]). Data sharing rates were low in both groups, with just two datasets shared in each of the intervention and control groups. The global movement towards open science has made significant gains with the development of numerous data sharing policies and tools. What remains to be established is an effective incentive that motivates researchers to take up such tools to share their data.

# 1. Introduction

There has been much recent debate and concern surrounding the transparency and integrity of health and medical research. We are living in a time where terms such as 'post-truth era' and 'reproducibility crisis' characterize the state of the worldwide scientific community, where even scientific truth has become a negotiable commodity. To address such a crisis, the meta-research community has, in collaboration with journals, universities, government bodies and other like-minded groups and organizations, developed an evolving framework for action to promote the integrity and transparency of health and medicine. Such a framework for action involves the implementation of regulations, standards, policies and tools that might generate and promote a movement towards open and transparent research. For instance, in 2015, Nosek and colleagues proposed the transparency and openness promotion (TOP) guidelines to assist journals and funders to adopt transparency and reproducibility policies [1–3]. Over 5000 journals, including *Science* and the group of *Springer Nature* journals and organizations, are signatories of these guidelines [2,4].

One element of this framework for action is the sharing of the underlying data from health and medical research. Such underlying data underpins all scientific claims that a study might make, as therein lies the evidence to support any claims. There are three main advantages to data sharing. Firstly, data sharing allows for the verification of results, as with access to data others can repeat analyses to verify the original results. Secondly, data sharing allows for checks for robustness of the results, such as data transformations and manipulations, to check if the results are robust to such changes. Thirdly, data sharing allows for new questions to be relatively quickly answered using data that is already available. Collecting data usually takes a significant amount of resources, so if other researchers can re-use existing data to answer their own questions, this should reduce research waste. A recent study in the United States found that the public trust of science was increased by open data, where 57% of US adults said they would trust scientific research findings more if researchers make their data publicly available [5]. Data sharing, then, is a key component of the movement towards science that is open, where scientific knowledge is easily accessible, replicable, verifiable, robust, and which contributes to new questions and findings. Journal data sharing policies have a crucial role to play in promoting data sharing [6].

Studies have demonstrated low rates of data sharing in health and medical research, with leading journals such as the *BMJ* having rates as low as 4.5% for articles published between 2009 and 2015 [7] and a rate of 0% for biomedical research articles published between 2000 and 2004 [8]. A recent update to this study, however, has shown an increase of data sharing in biomedical research articles published between 2015 and 2017 where 18% of papers discuss data availability [9].

In early 2016, the International Committee of Medical Journal Editors (ICMJE) put together a proposal outlining some requirements to help meet the mandating of clinical trial data sharing worldwide [3,10]. After receiving some criticism from researchers worldwide regarding the feasibility of the proposed requirements, in June 2017, the ICMJE released its revised requirements for data sharing statements of clinical trials, which only mandate the inclusion of data sharing statements but not data sharing [3,11]. Accordingly, despite the wide-scale calls for and benefits of data sharing, it is yet to become routine research policy and practice.

There are now many data repositories to openly and easily share and store scientific data (e.g. Figshare, Dryad). Yet the uptake of data sharing in health and medicine is still low. Naturally, not all health and medical data can be shared due to patient confidentiality; however, where data sharing is applicable, one barrier to uptake may be the lack of recognition for data sharing and hence a reward may be a useful incentive [12].

Incentives are defined here as rewards that are given to researchers if they participate in sharing their raw scientific data openly in a data repository. Data sharing requires effort and time on the part of researchers. Data need to be cleaned and prepared so that it can be understandable and useable by others. Uploading and storing data at a repository also requires time and resources. As such, it is anticipated that researchers might want recognition for their efforts to share data. Simple incentives might provide the change needed to increase data sharing in health and medical research. A systematic review showed that only one incentive has been used in health and medical research to motivate researchers to share their raw data: badges for data sharing [13].

Badges for code and data sharing were first adopted by the journal *Biostatistics* in 2009. In 2014, the Center for Open Science (USA) developed the Open Data Badge that motivated researchers to share their data with others [13]. In their observational study, Kidwell *et al.* [13], in collaboration with the journal *Psychological Science*, used Open Data Badges to reward researchers for sharing their data, which was related to an increased data sharing rate at *Psychological Science* (from 1.5% to 39%) [13].

A study published in 2017 found that while badges were associated with a rapid increase in data sharing between 2013 and 2015 at *Psychological Science*, in 2016 this trend stopped [14]. An observational study that examined the effect of badges at the journal *Biostatistics* found that badges were associated with increased data sharing of only 7.6% and had no effect on code sharing [15]. So, badges were related to a slight increase in data sharing, but had a limited efficacy. This study concluded that the best way to test the efficacy of badges was to conduct a randomized controlled trial (RCT) at a journal, thus providing the rationale for our study design. Previous studies that examined badges as incentives for data sharing were all observational, which are limited as they do not provide the evidence of causality because any observed changes may be due to confounding. As such, our study is the first experiment which tests the power of badges using the gold standard of research, a randomized controlled trial.

Our study setting was *BMJ Open* as in the worldwide health and medical publishing setting, *BMJ Open* has relatively strong policies to encourage data sharing. *BMJ Open* was the first medical journal to link datasets from its published medical journal articles to Dryad, an open repository [16]. Every *BMJ Open* research article must include a data sharing statement, even if it is to state that no data are available. The aim of this study was to examine if Open Data Badges increase data sharing rates among health and medical researchers that publish in *BMJ Open*.

# 2. Research design and methods

This study was a collaboration between *BMJ Open* and Queensland University of Technology (QUT). A low-risk ethics approval was provided by the Queensland University of Technology Human Research Ethics Committee (approval number: 1600001100). This study was a parallel group randomized controlled trial with two groups, control and intervention, with 80 research articles per group, with a total of 160 research articles.

## 2.1. Sample size calculation and power

A sample of 171 papers per group would have given us a 90% power to detect a difference in data sharing of 8% (average baseline data sharing rate at journals based on five published studies) versus 20% (a conservative halving of the previous badges study effect). This uses a two-sided 5% significance level. Due to unexpected slow recruitment, unfortunately the final sample was reduced to 80 papers per group, with a 60% power, which is quite low, to detect the 8% versus 20% difference in data sharing. However, if we wanted to detect the same effect size as the previous badges study, that is a 37.5% effect, 80 papers per group gives us a 99.7% power to detect the 8% versus 37.5% difference in data sharing.

# 3. Material and methods

## 3.1. Inclusion criteria

Papers were eligible for inclusion if:

— They were undergoing peer review at *BMJ Open*.

## 3.2. Exclusion criteria

Papers were excluded if:

— The paper's title contained the word 'protocol', as this meant it was likely to be a protocol rather than a results paper.
— They were meta-analyses or systematic reviews, as these papers often contain the data within the paper.
— They were case series, opinion pieces or some other publication type where there are no data.
— Any authors on the paper had a relationship with the QUT study team, in order not to bias their response.
— They were still under review at the time we assessed data sharing (see below).
— The contact author had already been approached to be part of the study.
— They were rejected after peer review.

A few papers that did meet the exclusion criteria were wrongly included and randomized, but were excluded before analysis.

## 3.3. Recruitment methods

*BMJ Open* included an opt-in button on the online submission system (ScholarOne) so that authors who were interested to participate could opt in. The QUT research team received author contact details after they opted in to be a part of this study upon submission of their paper to *BMJ Open* from 30 June 2017 to 16 March 2018.

Each potential paper was screened for eligibility and then added to a REDCap database [17]. REDCap provided a secure link for data transfer from the UK to Australia. The database recorded each study's title, study design, contact author, contact author's email, *BMJ Open* submission number and submission date.

A randomization list was created by the study statistician (Adrian Barnett) in R (www.r-project.org). It used 1:1 allocation in randomized blocks of size 4, 6 and 8. This blocking ensured roughly equal allocation over time. The randomization list was added to REDCap and papers were randomized by clicking a button after they were screened and had their basic details entered.

The peer review process created a delay between randomization and time to publication. At the time of recruitment in 2017, the median time to first decision at *BMJ Open* was 55 days, and manuscripts could have gone through two revisions. There was also a time between final acceptance and publication to allow authors to check the proofs of their article. We assessed the outcomes in November 2018. Any papers still under review at that time were excluded.

## 3.4. Treatment groups

The intervention group received an email (recruitment email included as supplementary material) which informed them about data sharing at *BMJ Open* as well as the treatment, an Open Data Badge in exchange for publicly sharing their data at a repository of their choice. The control group simply received an email (recruitment email included as electronic supplementary material) informing them about data sharing at *BMJ Open*, with no offer of an Open Data Badge or an incentive of any sort should they choose to publicly share their data at a repository of their choice.

# 4. Outcome measures

## 4.1. Primary outcome

— Data sharing rate: the number of papers sharing data divided by the total number of papers.

Data sharing was confirmed only after the data have been:

1. Directly available: deposited at a third-party site (e.g. Dryad, Figshare, Github, Kaggle).
2. Verified: confirmed by the study team as accessible data. Accessibility was verified by downloading the dataset and ensuring that it could be opened and that it contained data. We did not check to ensure that these data were relevant or complete as we did not have access to the articles' abstract or full text to verify those factors.

This definition of data sharing was derived from the Open Science Framework standard for earning an Open Data Badge [18].

A categorical data sharing outcome was made using the categories:

1. No data deposited at a public repository.
2. Data deposited at a public repository but data could not be verified due to embargoes and/or broken links.
3. Author(s) stated that data are available only after further applications (e.g. ethics), or other conditions (e.g. payment, travelling to host institution), or upon reasonable request. Or author(s) stated that data are available via a repository where access is provided only after a proposal to use data has been approved by an access committee, and within the terms of a data sharing agreement. Hence, data are available under controlled access.
4. Data deposited in the public repository without restriction.

Based on the above we also tabulated the data as 'Not available' (categories 1, 2), 'Potentially available' (category 3) and 'Directly available' (category 4).

*BMJ Open* staff sent through authors' data sharing statements once their articles were accepted for publication. The verification of data sharing was done independently by the two QUT investigators, with disagreements about which category a paper belongs in resolved by consensus. The verification was done blind to the treatment group; however, one investigator (Anisa Rowhani-Farid) might have been somewhat unblinded as she was involved with both recruiting participants (30 June 2017 to 16 March 2018) and verifying data sharing (December 2017 onwards), and hence may have known which group participants were in. The six-month delay between recruitment and data sharing verification should have reduced the possible recognition of authors.

## 4.2. Badge criteria

Using the Kidwell *et al.* [13] badge study as a precedent, the criteria for awarding papers with an Open Data Badge were as follows:

— The paper has been accepted for publication at *BMJ Open*.
— The paper has a clear, permanent link to an open repository where the data are stored.
— The data are easily downloadable (accessible), meaning that the data are able to be downloaded and opened up with no restrictions.

Thus, papers were only eligible for a badge if they provided direct access to data at a public repository, the 'Directly available' group (category 4).

Once data sharing verification for control and intervention groups were completed, the control group who made their data directly available also received a badge as a bonus.

Due to time and human resource constraints, the investigators did not apply for access to the data from the 'Potentially available', as this could have involved multiple lengthy ethical forms and application charges. *BMJ Open* also did not act as brokers to negotiate data access for researchers who declined to share their data.

# 5. Statistics methods

## 5.1. Primary outcome

We compared the data sharing rates for those who were (intervention group) and were not (control group) promised an Open Data Badge. We tabulated the numbers shared (yes/no) by treatment group (intervention/control) in a $2 \times 2$ table and used a Fisher's exact test as there were small cell sizes. We calculated the per cent shared in each treatment group, the difference between the groups and a 95% confidence interval of the difference.

## 5.2. Secondary outcomes

A potential secondary effect of badges is changes in what authors write in their data sharing statements. For example, the intervention might have prompted authors to be more detailed in explaining why they could or could not share their data. To test this, we collected the text in every 'Data sharing statement' and used a word frequency table to compare the two study groups. We first removed common words such as 'at', 'the', 'there' and 'their'. We also compared the average number of words per data sharing statement and calculated the difference and 95% confidence interval of the difference. We used a Poisson regression model with over-dispersion to examine the association between the intervention and the number of words in the data sharing statement.

## 5.3. Other data

To assess if the study groups were similar, we compared them in terms of: study type and publication status (accepted/rejected) using frequency tables.

We prospectively collected data on the amount of time needed by the QUT study team to verify the datasets.

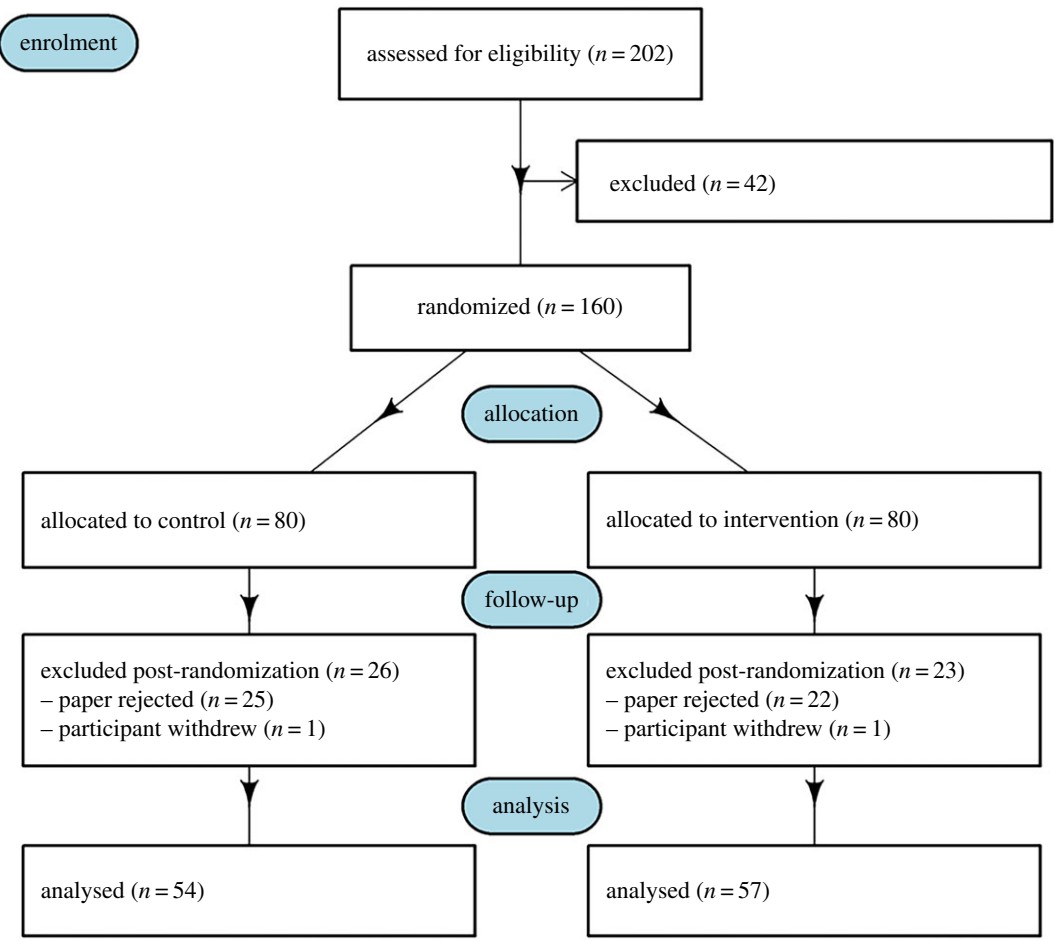

**Figure 1.** CONSORT flow chart for the RCT testing badges at *BMJ Open*.

A preliminary set of results was created using a scrambled treatment group in order to find problems with the data or methods before seeing the actual results. All analyses were made using the R software (v. 3.5.2).

# 6. Results

## 6.1. Participant recruitment

The CONSORT flow chart is in figure 1 [19]. Two hundred and two authors of papers consented to be a part of the study, 42 papers were excluded leaving 160 papers for randomization. Eighty papers were allocated per group, control and intervention. For the follow-up phase, 26 papers were excluded post randomization for the control group and 23 for the intervention group, which left 54 papers and 57 for the control and intervention group, respectively, for analysis.

## 6.2. Comparing the control and intervention groups at baseline

The study groups were comparable for all categorical variables (type of study, acceptance/rejection at *BMJ Open*, publication status and study withdrawal rate) with only some slight differences for the frequencies in the type of study category (table 1 and figure 2).

## 6.3. Primary outcome

The numbers of authors receiving a badge by treatment group—control and intervention—are shown in table 2. Papers in the control group who shared their data at a public repository were also offered an Open Data Badge after the trial ended. As such, 2 out of 54 papers in the control group received a

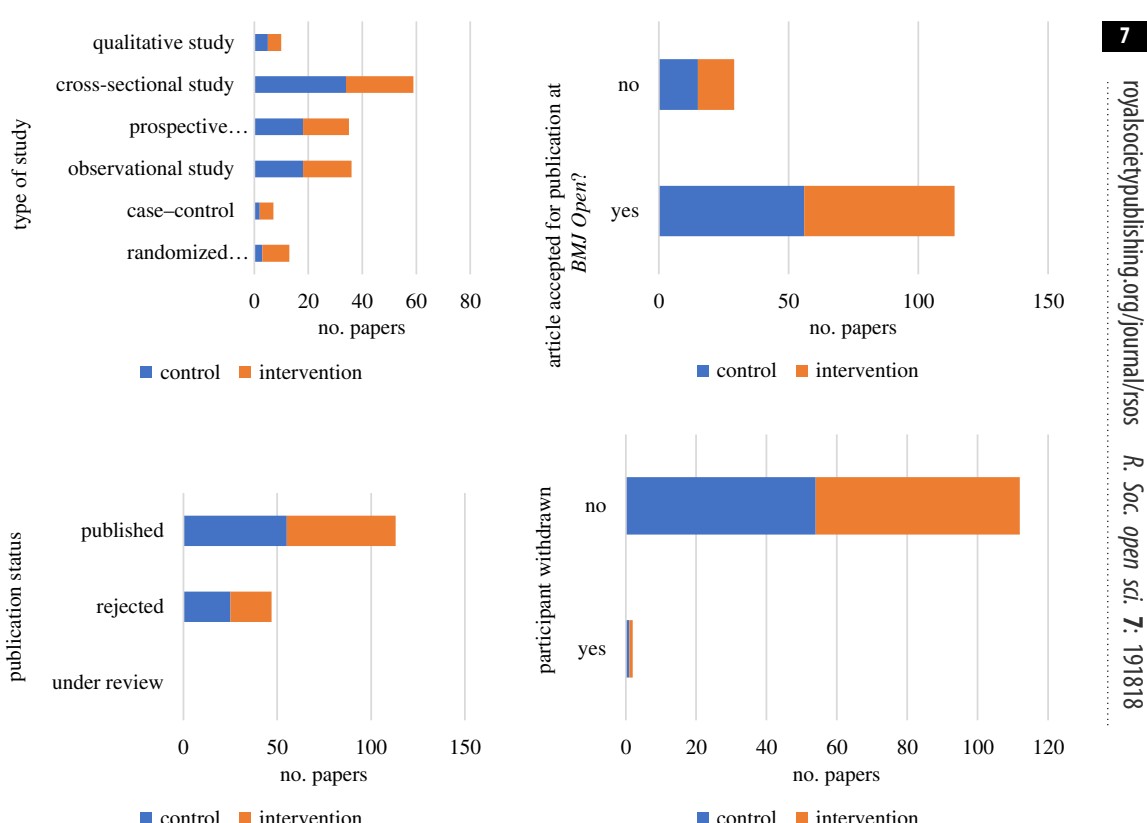

**Figure 2.** Charts of categorical variables comparing the control and intervention groups at baseline.

**Table 1.** Frequency table of categorical variables comparing the control and intervention groups at baseline. Some percentages do not add to 100% because of rounding.

| variable | category | control | | intervention | |
|---|---|---|---|---|---|
| | | *n* | % | *n* | % |
| type of study | randomized controlled trial | 3 | 4 | 10 | 12 |
| | case–control | 2 | 2 | 5 | 6 |
| | observational study | 18 | 22 | 18 | 22 |
| | prospective observational study (cohort study) | 18 | 22 | 17 | 21 |
| | cross-sectional study | 34 | 42 | 25 | 31 |
| | qualitative study | 5 | 6 | 5 | 6 |
| article accepted for publication at *BMJ Open*? | yes | 56 | 70 | 58 | 72 |
| | no | 15 | 19 | 14 | 18 |
| publication status | under review | 0 | 0 | 0 | 0 |
| | rejected | 25 | 31 | 22 | 28 |
| | published | 55 | 69 | 58 | 72 |
| participant withdrawn | yes | 1 | 1 | 1 | 1 |
| | no | 54 | 68 | 58 | 72 |

badge, 4% of papers, which is the same number as in the intervention group who were awarded a badge (2 out of 57 papers).

The odds of awarding badges are nearly equal in the intervention and control groups (OR=0.9, 95% CI [0.1, 9.0]). No statistically significant differences in awarding badges were found between intervention

**Table 2.** Numbers of papers receiving a badge by treatment group—control and intervention.

| | control | | intervention | |
|---|---|---|---|---|
| awarded a badge | *n* | % | *n* | % |
| yes | 2 | 4 | 2 | 4 |
| no | 52 | 96 | 55 | 96 |
| all | 54 | 100 | 57 | 100 |

**Table 3.** Numbers of papers per type of final data sharing statement by treatment group—control and intervention. Some percentages do not add to 100% because of rounding.

| | control | | intervention | |
|---|---|---|---|---|
| final data sharing statement | *n* | % | *n* | % |
| no additional data is available | 21 | 39 | 23 | 40 |
| data is available upon request | 30 | 56 | 32 | 56 |
| data is available at a third-party depository | 3 | 6 | 2 | 4 |
| all | 54 | 100 | 57 | 100 |

and control groups (Fisher's exact test, *p*-value = 1). Badges did not noticeably motivate researchers that publish in *BMJ Open* to publicly share their raw data as the odds ratio of awarding badges in the intervention group relative to the control is close to 1 (0.9). However, given that the confidence interval is wide (0.1 to 9.0), we cannot rule out an effect of badges that would have practical significance.

We had originally planned (in our pre-registered protocol) to test whether there was an interaction between the effect of badges (intervention or control arm) and study type; however, given the low number of papers that earned a badge, we did not examine this interaction.

The numbers of papers per type of final data sharing statement are shown in table 3, which has more categories than the binary badge (yes or no) variable. One researcher in the control group had their dataset under embargo until the article was published, which meant they could not receive a badge. The percentages per statement are comparable between treatment groups where 39% and 40% of papers in the control and intervention group, respectively, had no additional data available, 56% of papers in both groups had data available upon request, and 6% and 4% of papers in the control and intervention groups, respectively, had data available at a third-party repository.

## 6.4. Secondary outcomes

### 6.4.1. Words used in the final data sharing statements

To examine whether the intervention had an effect on the language used, we studied words commonly used in the final data sharing statements.

The difference in ranking in the top 20 words between the two groups is shown in figure 3. The top three words in order of ranking for both groups were 'data/dataset', 'available' and 'additional'.

When looking at the common three-word phrases between the two groups, 'no additional data' had the highest number of counts for both group with 14 counts per group, 'the corresponding author' had 13 counts and ranked second for the control group and had 12 counts and ranked third for the intervention group, and 'data are available' ranked third for the control group with 10 counts and ranked second with 13 counts for the intervention group. These findings indicate that the intervention did not have an effect on the language used in data sharing statements.

### 6.4.2. Number of words used in the final data sharing statements

We compared the number of words used in final data sharing statements to examine whether those in the intervention group tended to write longer data sharing statements. The box plot in figure 4

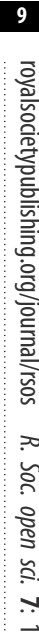

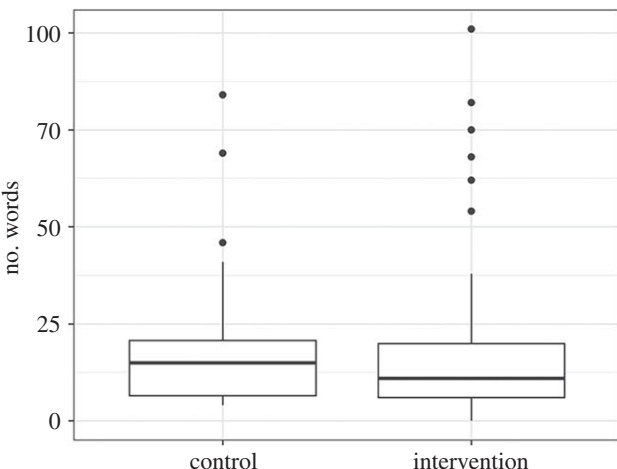

**Figure 3.** Plot showing the difference in ranking in the top 20 words between the treatment groups—control and intervention.

**Figure 4.** Boxplot of number of words used in final data sharing statements by treatment group—control and intervention.

shows comparable distributions for the number of words in data sharing statements between intervention and control.

There was no association between the intervention and the number of words. The mean rate ratio was 1.05 with a 95% confidence interval from 0.73 to 1.53.

### 6.4.3. Time needed to verify datasets

The mean time needed to check for open data by QUT study team was 3 min ($n = 8$, minimum = 1 min, maximum = 5 min).

# 7. Discussion

The findings of this RCT have demonstrated that Open Data Badges did not noticeably motivate researchers who publish with *BMJ Open* to publicly share their raw data (odds ratio 0.9, 95% CI: 0.1 to 9.0). These findings stand in contrast with those of Kidwell *et al.* [13] who found that Open Data Badges were associated with an increase in data sharing from 1.5% to 39% at *Psychological Science* [13]. A possible explanation behind these striking differences in data sharing rates is that health and medical data are often harder to share than psychological science data, which is largely due to issues around patient confidentiality and consent to share health and medical data publicly. Also, the reproducibility movement is more established in the field of psychology where it is becoming more of a cultural norm, so psychological science researchers might be more primed to share their data than health and medical researchers [20]. We found that the majority of health and medical researchers preferred to share their data upon request (56% of participants in both study groups). It seems sharing data via a controlled platform might receive greater uptake than that of publicly sharing data. As such, journals could consider providing platforms for controlled sharing of health and medical data for researchers who wish to share their data, but who have concerns about making their data public with unrestricted access.

It was surprising that badging did not work on an individual author level, particularly as the Kidwell *et al.* [13] study mentioned that '…*the mere fact that the journal engages authors with the possibility of promoting transparency by earning a badge may spur authors to act on their scientific values*'. [13]. In our email communication with an author of the Kidwell *et al.* [13] study about some possible explanations behind our study findings, a point that was raised was that in order for badging to work, a stronger signal from the journal was required, similar to how *Psychological Science* used signalling on articles, on the journal website, and in weekly emails announcing new articles or issues (Brian Nosek 2019, personal communication). Hence the approach used here of a single approach to authors may not have been sufficient to change behaviour. Studying the effect of this policy change would need a study that randomized at a journal rather than author level, but this would be more difficult and costly. Other areas of health have had some success with using 'bundles', where it is not just about one change but a raft of changes that occur at the same time, as a method of changing practice [21]. Perhaps badges could also be incorporated as a part of a data sharing encouragement 'bundle'.

Future RCTs could investigate alternative incentives for data sharing. One incentive that could be tested is funding. Funding organizations could offer additional funding as part of a larger grant to researchers who share their raw data. One such example of a funding body incentivizing data sharing is Berlin Institute of Health (BIH) and BIH QUEST Center rewarding scientists at Charité – Universitätsmedizin Berlin and the Max Delbrück Center for Molecular Medicine for disclosing the original data of their publications with a total sum of 200 000 euros to be divided among all those who share [22]. Journals could also offer funding as an incentive to sharing data by providing a discount in article processing charges if researchers shared their data.

While designing incentives for open data practices, it is important to consider the relatively high costs of data sharing versus low benefits [23]. Ioannidis [23] elaborates on these high costs as the 'risks' of replication or detection of error where a researcher spends a high amount of time cleaning and preparing data in order to share, only to potentially become more susceptible to criticism and replication attacks [23]. As such, we need to create a culture where sharing data and finding errors in analyses are normal and acceptable so as to minimize unnecessary attacks and criticism when shared data are not replicable [23].

## 7.1. Limitations

Due to the opt-in method of participant recruitment, the trial results could have overestimated the effect of badges, as researchers who are more sympathetic to data sharing might have been more likely to participate in the trial. However, we observed an overall low data sharing rate and no clear effect of badges in even this potentially more 'data-interested' group.

Our original study design was to randomize authors without individual consent from authors, as this would give a more realistic control group. The *BMJ Open* email to authors mentions the journal's

commitment to constantly improving the peer review system using research and offers authors an opt-out. However, our ethics committee disagreed with our approach and felt that individual consent was needed.

The estimated impact of badges might have been slightly reduced as articles must have their data available at the production phase in order to be eligible for a badge. One researcher had their dataset under embargo until the article was published, which meant they could not receive a badge. This is a logistical difficulty that journals need to consider if they decide to use badges.

# 8. Conclusion

The global movement towards open science has made recent significant gains in the development of numerous data sharing policies and tools. What remains to be established is an effective incentive that motivates researchers to take up such tools to share their data. An effective incentive has the potential to influence the scientific reward system so as to incorporate data sharing as a measure of high-quality research. Previous literature has shown that Open Data Badges were associated with motivating researchers to share their data; however, the findings of our trial demonstrate otherwise. Badges might still have the potential to change data sharing behaviour but the way in which they are introduced, for example with signalling or as a part of an encouragement 'bundle', could be associated with such a change.

Ethics. This study was a collaboration between *BMJ Open* and Queensland University of Technology (QUT). A low-risk ethics approval was provided by the Queensland University of Technology Human Research Ethics Committee (approval no: 1600001100).

Data accessibility. This study's anonymized dataset and code used to analysed the data are publicly available at a third-party repository and can be accessed via this permanent link: https://doi.org/10.5281/zenodo.3490012.

Competing interests. During the study, A.A. was Editor-in-Chief of *BMJ Open* and A.G.B. was on the Editorial and Statistical Board of *BMJ Open*. A.R.-F. has no competing interests.

Authors' contributions. A.R.-F. conceived of and designed the study, collected and interpreted data, drafted the first version of the manuscript and coordinated the study as a part of her PhD. A.A. conceived of and designed the study, collected and interpreted data and edited the manuscript. A.G.B. conceived of and designed the study, analysed and interpreted data, edited the manuscript and provided provided primary supervision for A.R.-F.'s PhD. All authors gave final approval for publication.

Funding. A.R.-F. was on QUT PhD and Write-Up scholarships and received in-kind support from the Institute of Health and Biomedical Innovation at QUT and *BMJ Open* during her internship there. A.G.B. was supported by Queensland University of Technology and the National Health and Medical Research Council grant no. APP1117784.

Acknowledgements. We would like to thank Michelle Allen (A.R.-F.'s PhD colleague) who contributed to conceiving of this study. We would like to acknowledge QUT for providing financial and in-kind support for the progress of this trial and *BMJ Open* for in-kind support while A.R.-F. was conducting her internship there to work on the trial.

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
