## [Reviewer comments · Royal Society Open Science]

Review History

RSOS-191818.R0 (Original submission)

Review form: Reviewer 1 (Andreas Schneck)

Is the manuscript scientifically sound in its present form?

Yes

Are the interpretations and conclusions justified by the results?

Yes

Is the language acceptable?

Yes

Do you have any ethical concerns with this paper?

No

Have you any concerns about statistical analyses in this paper?

No

Recommendation?

Major revision is needed (please make suggestions in comments)

Comments to the Author(s)

Dear Mrs. Rowhani-Farid, dear Mr. Aldcroft, dear Mr. Barnett,

Thank you for your well written and designed study that will contribute to the search for useful interventions to increase data-sharing along with publications. Nonetheless, I have remarks that should be addressed before publication.

Best wishes

Andreas Schneck

1. I suggest stressing the advantages of RCTs more strongly against observational studies. Your studies seem to be the first to vary an open data badge experimentally.
2. The methodological details of the paper in sections 1.3-1.5 (3.5 pages) could be shortened in the main text and reported in the supplemental material in more detail. Especially the bullet-points (i.e., in 1.4.2) decrease the readability a lot.
3. The opt-in procedure at the beginning of the RCT could, as the authors correctly note (p. 17 l. 20f.), imply selection bias. This however only concerns the external validity of the results and not the internal validity of the treatment, which is entirely unaffected. Because of the low baseline risk even in a "data-interested" subgroup as well as the absent overall effect of badges is, in my opinion, a strong point that badges seem not to work at all. In their limitations the authors may focus more on this argument instead of claiming the "over-conservatism" of their ethics committee.
4. Also, section 1.7.1 can be moved in the supplemental material.
5. In section 1.7.2 the reporting of the test-values should be oriented at a standard reporting guideline (e.g. of the American Psychological Association, 2010). E.g.: "The chances of sharing data are nearly equal in test and control group (OR = 0.9, 95% CI [0.1, 9.0]). No significant differences in data sharing were found between test and control group (Fisher's exact test, $p = 1$)".
6. I also suggest pruning the sentence "However, given that the confidence interval is wide (0.1 to 9.0), we cannot rule out a possible badge effect.". Given the confidence interval you are even not possible to rule out negative effects of badges. What you find is, in from my point of view, a very robust null effect of the badge intervention.
7. I am not convinced by the effects of badges on the secondary outcomes (1.7.3.). These effects are a little ad-hoc and are also not specified in the pre-registration. I instead suggest moving section 1.7.3. in the online appendix as robustness check showing, that also in other aspects of data sharing, nothing has happened after the badge intervention.
8. Perhaps it is fruitful to think about the incentive structure of open data practices in more detail: relatively high costs of data sharing (higher "risk" of replication/ detection of errors, a higher amount of time that has to be invested in data cleaning/ preparation) versus only low gains (intrinsic motivation, maybe slight reputation gains). Based on this incentive structure it is not surprising that badges do not work. This situation changes however if further costs were imposed for not sharing the data or new benefits (special mentioning on the homepage of the journal, etc.). For similar thoughts see also (Ioannidis, 2016).

Literature:

American Psychological Association. (2010). *Publication Manual of the American Psychological Association* (6th ed.). Washington, DC: American Psychological Association.

Ioannidis, J. P. A. (2016). Anticipating Consequences of Sharing Raw Data and Code and of Awarding Badges for Sharing. *Journal of Clinical Epidemiology*, 70, 258-260.

Review form: Reviewer 2 (Michèle Nuijten)

Is the manuscript scientifically sound in its present form?

Yes

Are the interpretations and conclusions justified by the results?

No

Is the language acceptable?

Yes

Do you have any ethical concerns with this paper?

No

Have you any concerns about statistical analyses in this paper?

Yes

Recommendation?

Major revision is needed (please make suggestions in comments)

Comments to the Author(s)

Review

The authors performed a randomized controlled trial to investigate if awarding Open Data Badges would increase data sharing rates in BMJ Open. I think it is great that the authors chose to do an RCT to investigate this. At the moment, a lot of meta-research is observational, which means that causal relationships can often not be established with certainty. This study fills that gap and shows the direction in which (I think) meta-research should go.

I think this is an interesting and important study, and I applaud the authors for preregistering it and sharing their data and materials. I do, however, think that several sections of the manuscript could be improved to increase clarity and avoid overstatements. I make some concrete suggestions below.

Signed,
Michèle Nuijten

Introduction

I feel that the argumentation in the introduction can be improved. First of all, I think the authors can more clearly indicate why data sharing is so incredibly important. It is good to hear that US adults would trust scientific research findings more if the data are available, but this is not the main reason why people should do it. In my view, sharing data has three main advantages that I would add to the introduction. First, data sharing allows verification of results. With the raw data, people can redo the analyses to see if they can obtain the same results as reported in the paper. Second, data sharing allows checks for robustness of the result. If people have access to the data, they can try to run slightly different analyses (include/remove outliers, try other data transformations, etc.), to see if the results are robust to such changes. Third, sometimes new questions can be answered with data that are already available. Collecting data usually takes a significant amount of resources, so if other researchers can reuse existing data to answer their own questions, this could reduce research waste.

A second point that might be worth mentioning in the introduction, is that it can take quite some time to prepare your data in such a way that it is understandable and useable for others. I think mentioning this may strengthen the authors' point that researchers want recognition for putting in the effort to share their data.

Third, I think the structure of the introduction could improve by switching some paragraphs around. I would suggest the following order: "One element of this framework for action is ..." [...] "in promoting data sharing (6)." New paragraph: "Studies have demonstrated low rates of

data sharing ...” [...] “18% of papers discuss data availability (11).” New paragraph: “In early 2016, ...” [...] “it is yet to become routine research policy and practice.”

Causal language

Throughout the paper, there are instances where the authors use causal language, where only correlations have been observed. For instance, on page 4 the authors write “In their observational study, Kidwell et al. (2016) used Open Data Badges to reward researchers for sharing their data which increased the data sharing rate at the journal *Psychological Science* from 1.5% to 39% (13).” From this study, it is not possible to conclude that the badges actually increased data sharing rate. This is merely a correlation, so a better way of phrasing this would be “In their observational study, Kidwell et al. (2016) used Open Data Badges to reward researchers for sharing their data which was related to an increased data sharing rate at the journal *Psychological Science* (from 1.5% to 39%) (13).”

Similar cases in which causal conclusions are implied where they are not justified are:

* “A study published in 2017 found that while badges rapidly increased data sharing between 2013

and 2015 at *Psychological Science*, ...”

* An observational study that examined the effect of badges at the journal *Biostatistics*, found that badges increased data sharing by only 7.6% and had no effect on code sharing (15).

* “So, badges worked, but had a limited efficacy.”

* “These findings contrast to those of Kidwell et al. (2016) who found open data badges increased data sharing from 1.5% to 39% at *Psychological Science* (13).”

* “A possible explanation behind these striking differences in badge effects [...]”

* “Previous literature has shown that open data badges were effective in motivating researchers to share their data”

A related case is the last sentence: “Badges might still have the potential to change data sharing behaviour but they must be introduced with signalling or as a part of an encouragement ‘bundle’.” I may have missed it, but this suggestion for an intervention also seems to be based on observational data and speculation.

Power

It is unfortunate that the authors were not able to obtain the statistical power they were aiming at. Their estimated power of 60% is very low, and it is important to mention this limitation clearly in the limitations sections. It is possible that the low power is an explanation for the lack of effect in the study (although the authors should be careful with such post-hoc explanations of the results, of course).

The 60% power was calculated based on an effect size that was half of what has been detected in previous studies. What would be the power of this study to detect the same effect as in previous studies? I think it would be interesting to mention this at the end of the paragraph about power.

Operationalization “data sharing”

The authors explicitly mention that the data had to be shared on a third-party site. What if the data were shared as supplemental materials in *BMJ Open* itself? Would that count as a third-party site as well? Or doesn't *BMJ Open* allow such supplemental materials anyway?

As a second point, the authors required that the data had to be “accessible”. This needs some further explanation. What does “accessible” mean exactly? Was it also checked whether these were data that were actually relevant for the research question? Or would anything that looked like data be counted as open data, regardless of its relevance/completeness?

Similarly, on page 8, the authors also require the data to be “easily downloadable”. What does this mean?

Tables

There seem to be some reproducibility issues with the tables in the paper. In Table 1, the percentages of Type of Study, Article accepted for publication at BMJ Open, and Participants withdrawn do not seem to add up to 100% in the different conditions. The same holds for Table 3, where the percentages in the control condition add up to 101%.

I also wonder if it might facilitate interpretation of the results if the data in Table 1 were (also) depicted in a figure. This may make it easier to see differences (or lack thereof) between the conditions.

I think the interpretation of Table 3 is somewhat confusing. Are these categories that BMJ Open uses or did the authors create them based on reading all statements? Furthermore, I find it a bit confusing that Table 3 shows that 3 papers in the control condition had data available at a third-party repository, whereas the conclusion above was that only 2 papers received a badge. Does that mean that 1 paper had data in a third-party repository that were not easily downloadable?

In Table 3, I think it would be clearer to include all 4 data sharing categories in the table, not just 3. Now, there are 6

Data statement analyses

As an additional analysis, the authors also looked at the differences in word type and number of words between data statements of the control and intervention condition. It is not entirely clear to me what such a quantitative analysis would tell us. Did the authors have any substantive reason to look at these characteristics of the statements? Similarly, I don't think Figure 2 adds a lot to the paper. I think the figure is not intuitive and I wonder what the message is that it should tell me.

Since I don't know why we would expect any difference in type of words or number of words of the data statement between the two conditions (and what difference specifically we would expect), I would like to suggest to remove these analyses out of the paper. If the authors feel that these analyses should stay, then I would like to see some justification in the paper on why this is interesting to look at, or what type of differences they expected.

Data and code

I think it's really great that the authors shared their code in RMarkdown format. I could access and run the code with only two minor issues. First, in the R file `BMJOpenBadges_R_2019-05-24_1042.r`, there are some weird symbols on line 1, which causes an error if you try to source it. Furthermore, the code gives an error when I'm trying to generate the CONSORT diagram, because I didn't have a folder called “figures”.

Minor remarks

* In the preregistration, the authors indicate that they plan to test if there is an interaction between intervention and study type on open data badges. I don't think I've seen this test in the paper. This is probably due to the (extremely) low number of papers that earned a badge, but I think it would be good if the authors would add a sentence to the manuscript stating why they didn't test the interaction anymore.

* You could consider adding “RCT” to your key words. In my view, this is an important and unique characteristic of your study.

* p. 3 line 16: missing space: key component?

- * p. 4: the authors state that data badges are the only incentive that has been used for data sharing, but what about journal requirements? Would that not also count as an incentive? For instance, the default policy of PLOS journals that data have to be shared? Or the data statement from BMJ itself? I also miss this option in the discussion where other incentives for data sharing are discussed. I would argue it could also be an interesting RCT to see if making data sharing mandatory has any effect?
- * p. 4 line 12: I may be mistaken, but Kidwell and colleagues did not actually hand out the badges, did they? The phrase "Kidwell et al. used Open Data Badges to reward research" implies that they did.
- * p. 4 line 54: "A sample of 171 papers per group would have given us a 90% power to detect a difference in data sharing of 8% (based on five published studies)", I'm not sure what the authors mean with "based on five published studies"?
- * p. 4 line 56: I'm surprised the authors calculated power based on a two-sided test. They had a clear one-sided hypothesis, so I'm wondering if a one-sided test wouldn't be justified here.
- * p. 5 line 44: "A few papers that did not meet the exclusion criteria were wrongly included ..." How many?
- * p. 7 line 37: could the authors provide a (ballpark) inter-rater reliability here? And I'm also curious to hear what caused the discrepancies.
- * p. 8 line 48-50: the authors mention something about contacting the authors of the studies they included. What did they have to contact them for? I don't see this mentioned anywhere else in the paper.
- * p. 9 line 7: "202 papers consented ...", I would rephrase this. Papers cannot consent to anything ;)
- * p. 11 line 7-9: missing word in "Papers in the control group who shared their data at a public repository were also offered [an Open Data Badge?] after the trial ended."
- * p. 15 line 23: typo: participants
- * p. 16: "One such example of a funding body incentivizing data sharing is Berlin Institute of Health (BIH) and BIH QUEST Center rewarding scientists at Charité - Universitätsmedizin Berlin and the Max Delbrück Center for Molecular Medicine for disclosing the original data of their publications with a total of 200,000 euros (22)." I think I'm misunderstanding this sentence. It seems to say that if you share your data, you get 200,000 euros. That surely can't be true? Maybe sharing the data is a requirement for getting a grant of 200,000 euros? Perhaps rephrase.
- * p. 16: "the trial results could have overestimated the effect of badges". This seems impossible given that the observed effect was zero.

Decision letter (RSOS-191818.R0)

12-Dec-2019

Dear Dr Rowhani-Farid,

The editors assigned to your paper ("Did awarding badges increase data sharing in BMJ Open? A randomized controlled trial") have now received comments from reviewers. We would like you to revise your paper in accordance with the referee and Associate Editor suggestions which can be found below (not including confidential reports to the Editor). Please note this decision does not guarantee eventual acceptance.

Please submit a copy of your revised paper before 04-Jan-2020. Please note that the revision deadline will expire at 00.00am on this date. If we do not hear from you within this time then it will be assumed that the paper has been withdrawn. In exceptional circumstances, extensions may be possible if agreed with the Editorial Office in advance. We do not allow multiple rounds of revision so we urge you to make every effort to fully address all of the comments at this stage. If deemed necessary by the Editors, your manuscript will be sent back to one or more of the

original reviewers for assessment. If the original reviewers are not available, we may invite new reviewers.

- Data accessibility

If you wish to submit your supporting data or code to Dryad (<http://datadryad.org/>), or modify your current submission to dryad, please use the following link:
<http://datadryad.org/submit?journalID=RSOS&manu=RSOS-191818>

- Competing interests

- Authors' contributions

AB carried out the molecular lab work, participated in data analysis, carried out sequence alignments, participated in the design of the study and drafted the manuscript; CD carried out

the statistical analyses; EF collected field data; GH conceived of the study, designed the study, coordinated the study and helped draft the manuscript. All authors gave final approval for publication.

- Acknowledgements

- Funding statement

Kind regards,
Anita Kristiansen
Editorial Coordinator
Royal Society Open Science
openscience@royalsociety.org

on behalf of Dr Denes Szucs (Associate Editor) and Essi Viding (Subject Editor)
openscience@royalsociety.org

Comments to Author:

Reviewers' Comments to Author:

Reviewer: 1

Comments to the Author(s)

Dear Mrs. Rowhani-Farid, dear Mr. Aldcroft, dear Mr. Barnett,
Thank you for your well written and designed study that will contribute to the search for useful interventions to increase data-sharing along with publications. Nonetheless, I have remarks that should be addressed before publication.
Best wishes
Andreas Schneck

1. I suggest stressing the advantages of RCTs more strongly against observational studies. Your studies seem to be the first to vary an open data badge experimentally.
2. The methodological details of the paper in sections 1.3-1.5 (3.5 pages) could be shortened in the main text and reported in the supplemental material in more detail. Especially the bullet-points (i.e., in 1.4.2) decrease the readability a lot.
3. The opt-in procedure at the beginning of the RCT could, as the authors correctly note (p. 17 l. 20f.), imply selection bias. This however only concerns the external validity of the results and not the internal validity of the treatment, which is entirely unaffected. Because of the low baseline risk even in a "data-interested" subgroup as well as the absent overall effect of badges is, in my opinion, a strong point that badges seem not to work at all. In their limitations the authors may focus more on this argument instead of claiming the "over-conservatism" of their ethics committee.
4. Also, section 1.7.1 can be moved in the supplemental material.
5. In section 1.7.2 the reporting of the test-values should be oriented at a standard reporting guideline (e.g. of the American Psychological Association, 2010). E.g.: "The chances of sharing data are nearly equal in test and control group (OR = 0.9, 95% CI [0.1, 9.0]). No significant differences in data sharing were found between test and control group (Fisher's exact test, $p = 1$)".

6. I also suggest pruning the sentence “However, given that the confidence interval is wide (0.1 to 9.0), we cannot rule out a possible badge effect.”. Given the confidence interval you are even not possible to rule out negative effects of badges. What you find is, in from my point of view, a very robust null effect of the badge intervention.

7. I am not convinced by the effects of badges on the secondary outcomes (1.7.3.). These effects are a little ad-hoc and are also not specified in the pre-registration. I instead suggest moving section 1.7.3. in the online appendix as robustness check showing, that also in other aspects of data sharing, nothing has happened after the badge intervention.

8. Perhaps it is fruitful to think about the incentive structure of open data practices in more detail: relatively high costs of data sharing (higher “risk” of replication/ detection of errors, a higher amount of time that has to be invested in data cleaning/ preparation) versus only low gains (intrinsic motivation, maybe slight reputation gains). Based on this incentive structure it is not surprising that badges do not work. This situation changes however if further costs were imposed for not sharing the data or new benefits (special mentioning on the homepage of the journal, etc.). For similar thoughts see also (Ioannidis, 2016).

Literature:

American Psychological Association. (2010). *Publication Manual of the American Psychological Association* (6th ed.). Washington, DC: American Psychological Association.

Ioannidis, J. P. A. (2016). Anticipating Consequences of Sharing Raw Data and Code and of Awarding Badges for Sharing. *Journal of Clinical Epidemiology*, 70, 258-260.

Reviewer: 2

Comments to the Author(s)

Review

The authors performed a randomized controlled trial to investigate if awarding Open Data Badges would increase data sharing rates in BMJ Open. I think it is great that the authors chose to do an RCT to investigate this. At the moment, a lot of meta-research is observational, which means that causal relationships can often not be established with certainty. This study fills that gap and shows the direction in which (I think) meta-research should go.

I think this is an interesting and important study, and I applaud the authors for preregistering it and sharing their data and materials. I do, however, think that several sections of the manuscript could be improved to increase clarity and avoid overstatements. I make some concrete suggestions below.

Signed,
Michèle Nuijten

Introduction

I feel that the argumentation in the introduction can be improved. First of all, I think the authors can more clearly indicate why data sharing is so incredibly important. It is good to hear that US adults would trust scientific research findings more if the data are available, but this is not the main reason why people should do it. In my view, sharing data has three main advantages that I would add to the introduction. First, data sharing allows verification of results. With the raw data, people can redo the analyses to see if they can obtain the same results as reported in the paper. Second, data sharing allows checks for robustness of the result. If people have access to the data, they can try to run slightly different analyses (include/remove outliers, try other data transformations, etc.), to see if the results are robust to such changes. Third, sometimes new questions can be answered with data that are already available. Collecting data usually takes a significant amount of resources, so if other researchers can reuse existing data to answer their own questions, this could reduce research waste.

A second point that might be worth mentioning in the introduction, is that it can take quite some time to prepare your data in such a way that it is understandable and useable for others. I think mentioning this may strengthen the authors' point that researchers want recognition for putting in the effort to share their data.

Third, I think the structure of the introduction could improve by switching some paragraphs around. I would suggest the following order: "One element of this framework for action is ..." [...] "in promoting data sharing (6)." New paragraph: "Studies have demonstrated low rates of data sharing ..." [...] "18% of papers discuss data availability (11)." New paragraph: "In early 2016, ..." [...] "it is yet to become routine research policy and practice."

Causal language

Throughout the paper, there are instances where the authors use causal language, where only correlations have been observed. For instance, on page 4 the authors write "In their observational study, Kidwell et al. (2016) used Open Data Badges to reward researchers for sharing their data which increased the data sharing rate at the journal *Psychological Science* from 1.5% to 39% (13)." From this study, it is not possible to conclude that the badges actually increased data sharing rate. This is merely a correlation, so a better way of phrasing this would be "In their observational study, Kidwell et al. (2016) used Open Data Badges to reward researchers for sharing their data which was related to an increased data sharing rate at the journal *Psychological Science* (from 1.5% to 39%) (13)."

Similar cases in which causal conclusions are implied where they are not justified are:

* "A study published in 2017 found that while badges rapidly increased data sharing between 2013 and 2015 at *Psychological Science*, ..."

* An observational study that examined the effect of badges at the journal *Biostatistics*, found that badges increased data sharing by only 7.6% and had no effect on code sharing (15).

* "So, badges worked, but had a limited efficacy."

* "These findings contrast to those of Kidwell et al. (2016) who found open data badges increased data sharing from 1.5% to 39% at *Psychological Science* (13)."

* "A possible explanation behind these striking differences in badge effects [...]"

* "Previous literature has shown that open data badges were effective in motivating researchers to share their data"

A related case is the last sentence: "Badges might still have the potential to change data sharing behaviour but they must be introduced with signalling or as a part of an encouragement 'bundle'." I may have missed it, but this suggestion for an intervention also seems to be based on observational data and speculation.

Power

It is unfortunate that the authors were not able to obtain the statistical power they were aiming at. Their estimated power of 60% is very low, and it is important to mention this limitation clearly in the limitations sections. It is possible that the low power is an explanation for the lack of effect in the study (although the authors should be careful with such post-hoc explanations of the results, of course).

The 60% power was calculated based on an effect size that was half of what has been detected in previous studies. What would be the power of this study to detect the same effect as in previous studies? I think it would be interesting to mention this at the end of the paragraph about power.

Operationalization “data sharing”

The authors explicitly mention that the data had to be shared on a third-party site. What if the data were shared as supplemental materials in BMJ Open itself? Would that count as a third-party site as well? Or doesn't BMJ Open allow such supplemental materials anyway?

As a second point, the authors required that the data had to be “accessible”. This needs some further explanation. What does “accessible” mean exactly? Was it also checked whether these were data that were actually relevant for the research question? Or would anything that looked like data be counted as open data, regardless of its relevance/completeness?

Similarly, on page 8, the authors also require the data to be “easily downloadable”. What does this mean?

Tables

There seem to be some reproducibility issues with the tables in the paper. In Table 1, the percentages of Type of Study, Article accepted for publication at BMJ Open, and Participants withdrawn do not seem to add up to 100% in the different conditions. The same holds for Table 3, where the percentages in the control condition add up to 101%.

I also wonder if it might facilitate interpretation of the results if the data in Table 1 were (also) depicted in a figure. This may make it easier to see differences (or lack thereof) between the conditions.

I think the interpretation of Table 3 is somewhat confusing. Are these categories that BMJ Open uses or did the authors create them based on reading all statements? Furthermore, I find it a bit confusing that Table 3 shows that 3 papers in the control condition had data available at a third-party repository, whereas the conclusion above was that only 2 papers received a badge. Does that mean that 1 paper had data in a third-party repository that were not easily downloadable?

In Table 3, I think it would be clearer to include all 4 data sharing categories in the table, not just 3. Now, there are 6

Data statement analyses

As an additional analysis, the authors also looked at the differences in word type and number of words between data statements of the control and intervention condition. It is not entirely clear to me what such a quantitative analysis would tell us. Did the authors have any substantive reason to look at these characteristics of the statements? Similarly, I don't think Figure 2 adds a lot to the paper. I think the figure is not intuitive and I wonder what the message is that it should tell me.

Since I don't know why we would expect any difference in type of words or number of words of the data statement between the two conditions (and what difference specifically we would expect), I would like to suggest to remove these analyses out of the paper. If the authors feel that these analyses should stay, then I would like to see some justification in the paper on why this is interesting to look at, or what type of differences they expected.

Data and code

I think it's really great that the authors shared their code in RMarkdown format. I could access and run the code with only two minor issues. First, in the R file `BMJOpenBadges_R_2019-05-24_1042.r`, there are some weird symbols on line 1, which causes an error if you try to source it. Furthermore, the code gives an error when I'm trying to generate the CONSORT diagram, because I didn't have a folder called “figures”.

Minor remarks

- * In the preregistration, the authors indicate that they plan to test if there is an interaction between intervention and study type on open data badges. I don't think I've seen this test in the paper. This is probably due to the (extremely) low number of papers that earned a badge, but I think it would be good if the authors would add a sentence to the manuscript stating why they didn't test the interaction anymore.
- * You could consider adding "RCT" to your key words. In my view, this is an important and unique characteristic of your study.
- * p. 3 line 16: missing space: key component?
- * p. 4: the authors state that data badges are the only incentive that has been used for data sharing, but what about journal requirements? Would that not also count as an incentive? For instance, the default policy of PLOS journals that data have to be shared? Or the data statement from BMJ itself? I also miss this option in the discussion where other incentives for data sharing are discussed. I would argue it could also be an interesting RCT to see if making data sharing mandatory has any effect?
- * p. 4 line 12: I may be mistaken, but Kidwell and colleagues did not actually hand out the badges, did they? The phrase "Kidwell et al. used Open Data Badges to reward research" implies that they did.
- * p. 4 line 54: "A sample of 171 papers per group would have given us a 90% power to detect a difference in data sharing of 8% (based on five published studies)", I'm not sure what the authors mean with "based on five published studies"?
- * p. 4 line 56: I'm surprised the authors calculated power based on a two-sided test. They had a clear one-sided hypothesis, so I'm wondering if a one-sided test wouldn't be justified here.
- * p. 5 line 44: "A few papers that did not meet the exclusion criteria were wrongly included ..." How many?
- * p. 7 line 37: could the authors provide a (ballpark) inter-rater reliability here? And I'm also curious to hear what caused the discrepancies.
- * p. 8 line 48-50: the authors mention something about contacting the authors of the studies they included. What did they have to contact them for? I don't see this mentioned anywhere else in the paper.
- * p. 9 line 7: "202 papers consented ...", I would rephrase this. Papers cannot consent to anything ;)
- * p. 11 line 7-9: missing word in "Papers in the control group who shared their data at a public repository were also offered [an Open Data Badge?] after the trial ended."
- * p. 15 line 23: typo: participants
- * p. 16: "One such example of a funding body incentivizing data sharing is Berlin Institute of Health (BIH) and BIH QUEST Center rewarding scientists at Charité - Universitätsmedizin Berlin and the Max Delbrück Center for Molecular Medicine for disclosing the original data of their publications with a total of 200,000 euros (22)." I think I'm misunderstanding this sentence. It seems to say that if you share your data, you get 200,000 euros. That surely can't be true? Maybe sharing the data is a requirement for getting a grant of 200,000 euros? Perhaps rephrase.
- * p. 16: "the trial results could have overestimated the effect of badges". This seems impossible given that the observed effect was zero.

Author's Response to Decision Letter for (RSOS-191818.R0)

See Appendix A.

RSOS-191818.R1 (Revision)

Review form: Reviewer 1 (Andreas Schneck)

Is the manuscript scientifically sound in its present form?

Yes

Are the interpretations and conclusions justified by the results?

Yes

Is the language acceptable?

Yes

Do you have any ethical concerns with this paper?

No

Have you any concerns about statistical analyses in this paper?

No

Recommendation?

Accept with minor revision (please list in comments)

Comments to the Author(s)

Dear Authors,

Thank you very much for addressing my remarks and your efforts to revise the manuscript. I still have some issues that should be addressed before publication:

Major remarks:

1. I am still not convinced by the hypothesis concerning the secondary outcome; why should this be the case from a theoretical perspective? There should be at least some arguments why the used terminology in the data availability statement is expected to differ.
2. In the abstract, please report the OR in the same reporting style as in the paper.

Minor remarks:

1. The numbering of the paragraphs all starts with 1, is this intentional?
2. p.17 l.18 consider the term reanalysis instead of "can repeat analyses to"
3. p.23 l.36 change "the only number eligible for a badge is number 4" to "papers are only eligible if they provide direct access to the data (category 4).
4. The paper may need some format editing as some captions are numbered and some are not (e.g. "other outcome" on p.24 l.20). Also, the terminology should be checked as "other outcomes" are also referred to as "secondary outcomes" (p.29 l.14).

Review form: Reviewer 2 (Michèle Nuijten)

Is the manuscript scientifically sound in its present form?

Yes

Are the interpretations and conclusions justified by the results?

Yes

Is the language acceptable?

Yes

Do you have any ethical concerns with this paper?

No

Have you any concerns about statistical analyses in this paper?

No

Recommendation?

Accept with minor revision (please list in comments)

Comments to the Author(s)

I would like to thank the authors for their responses to both my comments and the ones of the other reviewer. I only have a couple of minor points, but other than that, I am looking forward to see this paper in print.

Signed,

Michèle Nuijten

I first would like to say that I agree with the authors' response to reviewer 1 that they decided to keep the methodological details in the paper. These details are important to have in order to be able to interpret the findings.

Next, I would like to clarify my point about Table 3 and the different categorizations of open data. I appreciate the authors' wish to keep the categories they developed in previous studies. I think I mixed up the categories of data sharing with the categories of data sharing *statements*. That said, the reason I brought it up in the first place, is that several different sets of classifications are used throughout the paper that are not all used in presenting the results.

First, the authors introduce four different categories:

A categorical data sharing outcome was made using the categories:

1. No data deposited at a public repository
2. Data deposited at a public repository but data could not be verified due to embargoes and/or broken links
3. Author(s) stated that data are available only after further applications (e.g. ethics), or other conditions (e.g. payment, travelling to host institution), or upon reasonable request. Or author(s) stated that data are available via a repository where access is provided only after a proposal to use data has been approved by an access committee, and within the terms of a data sharing agreement. Hence, data are available under controlled access.
4. Data deposited in public repository without restriction

Next, they summarize these four categories in three other categories:

Based on the above we also tabulated the data as

1. 'Not available' (categories 1, 2),
2. 'Potentially available' (category 3) and
3. 'Directly available' (category 4).

(As an aside, if I understand the text in the manuscript correctly, the authors would consider a broken link "not available", not as "potentially available" as they state in their response.)

These three categories are then further classified in:

1. Open Data Badge (only category 3)
2. No Open Data Badge (all other categories)

Unless I overlooked it, there are only results available for the final, dichotomous categorization: whether papers did or did not earn a Data Badge. I think that instead of presenting different sets of classifications, it might be clearer if the authors would present their dichotomization first, and then explain which situations would be eligible for a badge and which wouldn't. Otherwise, the reader might still be expecting the results for the four or three different categories that were presented earlier in the paper. It is of course possible that only I experienced this confusion, in which case incorporating this suggestion is not necessary.

Another point I made in my first review, was that it was unclear to me why the authors did a word count analysis in the data statements. In their response (and in the paper), the authors explain they were interested in seeing whether there was a difference in the language used between conditions. I see that reviewer 1 was also not convinced by the relevance of these analysis (see point 7 in his review), which signals to me that the rationale of this analysis is not sufficiently explained. If the authors are adamant on keeping this section in the paper, I would suggest adding some additional explanation in which the authors give some examples of the possible changes in language they expected to find (e.g., perhaps the authors expected that the intervention caused people to be more detailed in explaining why they couldn't share the data). Such examples may help convince the reader that these secondary analyses can give interesting insights in potential side-effects of giving out badges.

Then, the weird symbols in the R script were not in the Markdown file, but in the R file in the data folder (BMJOpenBadges_R_2019-05-24_1042.r). On line 1, it reads: `i>¿#Clear existing data and graphics.`

Finally, some additional comments about the minor remarks:

* p. 4 line 12: I may be mistaken, but Kidwell and colleagues did not actually hand out the badges, did they? The phrase "Kidwell et al. used Open Data Badges to reward research" implies that they did.

RESPONSE: Yes, they did give out badges (electronic badges on papers).

Reply to the authors: What I meant was that the paper implies that Kidwell and colleagues personally handed out badges, whereas the actual badges were awarded by the journal Psychological Science. Kidwell et al. just recorded whether a paper did or did not receive a badge.

Decision letter (RSOS-191818.R1)

25-Feb-2020

Dear Dr Rowhani-Farid:

On behalf of the Editors, I am pleased to inform you that your Manuscript RSOS-191818.R1 entitled "Did awarding badges increase data sharing in BMJ Open? A randomized controlled trial" has been accepted for publication in Royal Society Open Science subject to minor revision in accordance with the referee suggestions. Please find the referees' comments at the end of this email.

The reviewers and Subject Editor have recommended publication, but also suggest some minor revisions to your manuscript. Therefore, I invite you to respond to the comments and revise your manuscript.

- Ethics statement

- Data accessibility

<http://datadryad.org/submit?journalID=RSOS&manu=RSOS-191818.R1>

- Competing interests

- Authors' contributions

- Acknowledgements

- Funding statement

Because the schedule for publication is very tight, it is a condition of publication that you submit the revised version of your manuscript before 05-Mar-2020. Please note that the revision deadline

will expire at 00.00am on this date. If you do not think you will be able to meet this date please let me know immediately.

on behalf of Dr Denes Szucs (Associate Editor) and Essi Viding (Subject Editor)
openscience@royalsociety.org

Associate Editor Comments to Author (Dr Denes Szucs):

Associate Editor: 1

Comments to the Author:

Please revise all points raised by the reviewers.

Reviewer comments to Author:

Reviewer: 1

Comments to the Author(s)

Dear Authors,

Thank you very much for addressing my remarks and your efforts to revise the manuscript. I still have some issues that should be addressed before publication:

Major remarks:

1. I am still not convinced by the hypothesis concerning the secondary outcome; why should this be the case from a theoretical perspective? There should be at least some arguments why the used terminology in the data availability statement is expected to differ.
2. In the abstract, please report the OR in the same reporting style as in the paper.

Minor remarks:

1. The numbering of the paragraphs all starts with 1, is this intentional?
2. p.17 l.18 consider the term reanalysis instead of "can repeat analyses to"
3. p.23 l.36 change "the only number eligible for a badge is number 4" to "papers are only eligible if they provide direct access to the data (category 4).
4. The paper may need some format editing as some captions are numbered and some are not (e.g. "other outcome" on p.24 l.20). Also, the terminology should be checked as "other outcomes" are also referred to as "secondary outcomes" (p.29 l.14).

Reviewer: 2

Comments to the Author(s)

I would like to thank the authors for their responses to both my comments and the ones of the other reviewer. I only have a couple of minor points, but other than that, I am looking forward to see this paper in print.

Signed,

Michèle Nuijten

I first would like to say that I agree with the authors' response to reviewer 1 that they decided to keep the methodological details in the paper. These details are important to have in order to be able to interpret the findings.

Next, I would like to clarify my point about Table 3 and the different categorizations of open data. I appreciate the authors' wish to keep the categories they developed in previous studies. I think I mixed up the categories of data sharing with the categories of data sharing *statements*. That said, the reason I brought it up in the first place, is that several different sets of classifications are used throughout the paper that are not all used in presenting the results.

First, the authors introduce four different categories:

A categorical data sharing outcome was made using the categories:

1. No data deposited at a public repository
2. Data deposited at a public repository but data could not be verified due to embargoes and/or broken links

3. Author(s) stated that data are available only after further applications (e.g. ethics), or other conditions (e.g. payment, travelling to host institution), or upon reasonable request. Or author(s) stated that data are available via a repository where access is provided only after a proposal to use data has been approved by an access committee, and within the terms of a data sharing agreement. Hence, data are available under controlled access.
4. Data deposited in public repository without restriction

Next, they summarize these four categories in three other categories:

Based on the above we also tabulated the data as

1. `Not available' (categories 1, 2),
2. `Potentially available' (category 3) and
3. `Directly available' (category 4).

(As an aside, if I understand the text in the manuscript correctly, the authors would consider a broken link “not available”, not as “potentially available” as they state in their response.)

These three categories are then further classified in:

1. Open Data Badge (only category 3)
2. No Open Data Badge (all other categories)

Unless I overlooked it, there are only results available for the final, dichotomous categorization: whether papers did or did not earn a Data Badge. I think that instead of presenting different sets of classifications, it might be clearer if the authors would present their dichotomization first, and then explain which situations would be eligible for a badge and which wouldn't. Otherwise, the reader might still be expecting the results for the four or three different categories that were presented earlier in the paper. It is of course possible that only I experienced this confusion, in which case incorporating this suggestion is not necessary.

Another point I made in my first review, was that it was unclear to me why the authors did a word count analysis in the data statements. In their response (and in the paper), the authors explain they were interested in seeing whether there was a difference in the language used between conditions. I see that reviewer 1 was also not convinced by the relevance of these analysis (see point 7 in his review), which signals to me that the rationale of this analysis is not sufficiently explained. If the authors are adamant on keeping this section in the paper, I would suggest adding some additional explanation in which the authors give some examples of the possible changes in language they expected to find (e.g., perhaps the authors expected that the intervention caused people to be more detailed in explaining why they couldn't share the data). Such examples may help convince the reader that these secondary analyses can give interesting insights in potential side-effects of giving out badges.

Then, the weird symbols in the R script were not in the Markdown file, but in the R file in the data folder (BMJOpenBadges_R_2019-05-24_1042.r). On line 1, it reads: `ï»¿#Clear existing data and graphics.`

Finally, some additional comments about the minor remarks:

* p. 4 line 12: I may be mistaken, but Kidwell and colleagues did not actually hand out the badges, did they? The phrase “Kidwell et al. used Open Data Badges to reward research” implies that they did.

RESPONSE: Yes, they did give out badges (electronic badges on papers).

Reply to the authors: What I meant was that the paper implies that Kidwell and colleagues personally handed out badges, whereas the actual badges were awarded by the journal Psychological Science. Kidwell et al. just recorded whether a paper did or did not receive a badge.

Author's Response to Decision Letter for (RSOS-191818.R1)

See Appendix B.

Decision letter (RSOS-191818.R2)

27-Feb-2020

Dear Dr Rowhani-Farid,

It is a pleasure to accept your manuscript entitled "Did awarding badges increase data sharing in BMJ Open? A randomized controlled trial" in its current form for publication in Royal Society Open Science.

on behalf of Dr Denes Szucs (Associate Editor) and Prof Essi Viding (Subject Editor)
openscience@royalsociety.org

Appendix A

Dear Anita, Andreas and Michèle,

Thank you for your peer review, we greatly appreciate it. Find below our response to the editor and referees in blue.

Kind regards,
Anisa, Adrian A and Adrian B

12-Dec-2019

Dear Dr Rowhani-Farid,

The editors assigned to your paper ("Did awarding badges increase data sharing in BMJ Open? A randomized controlled trial") have now received comments from reviewers. We would like you to revise your paper in accordance with the referee and Associate Editor suggestions which can be found below (not including confidential reports to the Editor). Please note this decision does not guarantee eventual acceptance.

Please submit a copy of your revised paper before 04-Jan-2020. Please note that the revision deadline will expire at 00.00am on this date. If we do not hear from you within this time then it will be assumed that the paper has been withdrawn. In exceptional circumstances, extensions may be possible if agreed with the Editorial Office in advance. We do not allow multiple rounds of revision so we urge you to make every effort to fully address all of the comments at this stage. If deemed necessary by the Editors, your manuscript will be sent back to one or more of the original reviewers for assessment. If the original reviewers are not available, we may invite new reviewers.

If your study uses humans or animals please include details of the ethical approval received,

including the name of the committee that granted approval. For human studies please also detail whether informed consent was obtained. For field studies on animals please include details of all permissions, licences and/or approvals granted to carry out the fieldwork.

RESPONSE: Added this: This study was a collaboration between *BMJ Open* and Queensland University of Technology (QUT). A low-risk ethics approval was provided by the Queensland University of Technology Human Research Ethics Committee (approval number: 1600001100).

- Data accessibility

RESPONSE: Already had this.

<http://datadryad.org/submit?journalID=RSOS&manu=RSOS-191818>.

RESPONSE: We have uploaded our data and code at Zenodo.

- Competing interests

RESPONSE: Added this: All authors have no competing interests to declare.

- Authors' contributions

RESPONSE: Added this: ARF conceived of and designed the study, collected and interpreted data, drafted the first version of the manuscript and coordinated the study as a part of her PhD.

AA conceived of and designed the study, collected and interpreted data, and edited the manuscript.

AB conceived of and designed the study, analysed and interpreted data, edited the manuscript, provided provided primary supervision for ARFs PhD. All authors gave final approval for publication.

- Acknowledgements

RESPONSE: Added this: We would like to thank Michelle Allen (ARF's PhD colleague) who contributed to conceiving of this study. We would like to acknowledge *QUT* for providing financial and in-kind support for the progress of this trial and *BMJ Open* for in-kind support while ARF was conducting her internship there to work on the trial.

- Funding statement

RESPONSE: Added this: ARF was on *QUT* scholarships and received in-kind support from the Institute of Health and Biomedical Innovation at *QUT* and *BMJ Open* during her internship there. AGB was supported by Queensland University of Technology and the National Health and Medical Research Council grant number APP1117784.

Kind regards,

Anita Kristiansen

Editorial Coordinator

on behalf of Dr Denes Szucs (Associate Editor) and Essi Viding (Subject Editor)

Comments to Author:

Reviewers' Comments to Author:

Reviewer: 1

Comments to the Author(s)

Dear Mrs. Rowhani-Farid, dear Mr. Aldcroft, dear Mr. Barnett,

Thank you for your well written and designed study that will contribute to the search for useful interventions to increase data-sharing along with publications. Nonetheless, I have remarks that should be addressed before publication.

Best wishes

Andreas Schneck

1. I suggest stressing the advantages of RCTs more strongly against observational studies. Your studies seem to be the first to vary an open data badge experimentally.

RESPONSE: We have added a few sentences on page 4-5: Previous studies that examined badges as incentives for data sharing were all observational, which are limited as they do not provide evidence of causality because any observed changes may be due to confounding. As such, our study is the first experiment which tests the power of badges using the gold standard of research, a randomized controlled trial.

2. The methodological details of the paper in sections 1.3-1.5 (3.5 pages) could be shortened in the main text and reported in the supplemental material in more detail. Especially the bullet-points (i.e., in 1.4.2) decrease the readability a lot.

RESPONSE: We agree that it is long; however, we feel these details, particularly the exclusion criteria (1.4.2), are important to keep in the paper. We have chosen to keep these details in so that others who might wish to reproduce our trial at another journal, have access to our methodological details.

3. The opt-in procedure at the beginning of the RCT could, as the authors correctly note (p. 17 l. 20f.), imply selection bias. This however only concerns the external validity of the results and not the internal validity of the treatment, which is entirely unaffected. Because of the low baseline risk even in a “data-interested” subgroup as well as the absent overall effect of badges is, in my opinion, a strong point that badges seem not to work at all. In their limitations the authors may focus more on this argument instead of claiming the “over-conservatism” of their ethics committee.

RESPONSE: Added in a sentence on page 18: However, we observed an overall low data sharing rate and no clear effect of badges in even this potentially more “data-interested” group.

4. Also, section 1.7.1 can be moved in the supplemental material.

RESPONSE: This is our main summary results table and we wish to keep it in the main results of our paper instead of as supplemental material.

5. In section 1.7.2 the reporting of the test-values should be oriented at a standard reporting guideline (e.g. of the American Psychological Association, 2010). E.g.: “The chances of sharing data are nearly equal in test and control group (OR = 0.9, 95% CI [0.1, 9.0]). No significant differences in data sharing were found between test and control group (Fisher’s exact test, $p = 1$)”.

RESPONSE: Adjusted the wording for this paragraph on page 13: The odds of awarding badges are nearly equal in the intervention and control groups (OR = 0.9, 95% CI [0.1, 9.0]). No statistically significant differences in awarding badges were found between intervention and control groups (Fisher’s exact test, p -value = 1).

6. I also suggest pruning the sentence “However, given that the confidence interval is wide (0.1 to 9.0), we cannot rule out a possible badge effect.”. Given the confidence interval you are even not possible to rule out negative effects of badges. What you find is, in from my point of view, a very robust null effect of the badge intervention.

RESPONSE: We have rephrased the sentence on page 13 to read: However, given that the confidence interval is wide (0.1 to 9.0), we cannot rule out an effect of badges that would have practical significance.

7. I am not convinced by the effects of badges on the secondary outcomes (1.7.3.). These effects are a little ad-hoc and are also not specified in the pre-registration. I instead suggest moving section 1.7.3. in the online appendix as robustness check showing, that also in other aspects of data sharing, nothing has happened after the badge intervention.

RESPONSE: We have decided to keep these outcomes in our paper as we did outline these secondary outcomes in our protocol that was pre-registered (<https://osf.io/unxa3/>).

8. Perhaps it is fruitful to think about the incentive structure of open data practices in more detail: relatively high costs of data sharing (higher “risk” of replication/ detection of errors, a higher amount of time that has to be invested in data cleaning/ preparation) versus only low gains (intrinsic motivation, maybe slight reputation gains). Based on this incentive structure it is not surprising that badges do not work. This situation changes however if further costs were imposed for not sharing the data or new benefits (special mentioning on the homepage of the journal, etc.). For similar thoughts see also (Ioannidis, 2016).

RESPONSE: Added a paragraph about these concepts on page 18: While designing incentives for open data practices, it is important to consider the relatively high costs of data sharing versus low benefits (23). Ioannidis (2016) elaborates on these high costs as the “risks” of replication or detection of error where a researcher spends a high amount of time cleaning and preparing data in order to share, only to potentially become more susceptible to criticism and replication attacks (23). As such, we need to create a culture where sharing data and finding errors in analyses are normal and acceptable so as to minimize unnecessary attacks and criticism when shared data are not replicable (23).

Literature:

American Psychological Association. (2010). *Publication Manual of the American Psychological Association* (6th ed.). Washington, DC: American Psychological Association.

Ioannidis, J. P. A. (2016). Anticipating Consequences of Sharing Raw Data and Code and of Awarding Badges for Sharing. *Journal of Clinical Epidemiology*, 70, 258-260.

Reviewer: 2

Comments to the Author(s)

Review

The authors performed a randomized controlled trial to investigate if awarding Open Data Badges would increase data sharing rates in BMJ Open. I think it is great that the authors chose to do an RCT to investigate this. At the moment, a lot of meta-research is observational, which means that causal relationships can often not be established with certainty. This study fills that gap and shows the direction in which (I think) meta-research should go.

I think this is an interesting and important study, and I applaud the authors for preregistering it and sharing their data and materials. I do, however, think that several sections of the manuscript could be improved to increase clarity and avoid overstatements. I make some concrete suggestions below.

Signed,
Michèle Nuijten

Introduction

I feel that the argumentation in the introduction can be improved. First of all, I think the authors can more clearly indicate why data sharing is so incredibly important. It is good to hear that US adults would trust scientific research findings more if the data are available, but this is not the main reason why people should do it. In my view, sharing data has three main advantages that I would add to the introduction. First, data sharing allows verification of results. With the raw data, people can redo the analyses to see if they can obtain the same results as reported in the paper. Second, data sharing allows checks for robustness of the result. If people have access to the data, they can try to run slightly different analyses (include/remove outliers, try other data

transformations, etc.), to see if the results are robust to such changes. Third, sometimes new questions can be answered with data that are already available. Collecting data usually takes a significant amount of resources, so if other researchers can reuse existing data to answer their own questions, this could reduce research waste.

RESPONSE: Added in a paragraph on page 3 about the three main advantages for data sharing: There are three main advantages to data sharing. Firstly, data sharing allows for the verification of results, as with access to data others can repeat analyses to verify the original results. Secondly, data sharing allows for checks for robustness of the results, such as data transformations and manipulations, to check if the results are robust to such changes. Thirdly, data sharing allows for new questions to be relatively quickly answered using data that is already available. Collecting data usually takes a significant amount of resources, so if other researchers can reuse existing data to answer their own questions, this should reduce research waste. Adjusted the wording in the paragraph below this one to reflect these changes: Data sharing, then, is a key component of the movement towards science that is open, where scientific knowledge is easily accessible, replicable, verifiable, robust, and which contributes to new questions and findings.

A second point that might be worth mentioning in the introduction, is that it can take quite some time to prepare your data in such a way that it is understandable and useable for others. I think mentioning this may strengthen the authors' point that researchers want recognition for putting in the effort to share their data.

RESPONSE: Added a paragraph on page 4 about this: Data sharing requires effort and time on the part of researchers. Data need to be cleaned and prepared so that it can be understandable and useable by others. Uploading and storing data at a repository also requires time and resources. As such, it is anticipated that researchers might want recognition for their efforts to share data.

Third, I think the structure of the introduction could improve by switching some paragraphs around. I would suggest the following order: "One element of this framework for action is ..." [...] "in promoting data sharing (6)." New paragraph: "Studies have demonstrated low rates of data sharing ..." [...] "18% of papers discuss data availability (11)." New paragraph: "In early 2016, ..." [...] "it is yet to become routine research policy and practice."

RESPONSE: Switched those paragraphs around on page 3.

Causal language

Throughout the paper, there are instances where the authors use causal language, where only correlations have been observed. For instance, on page 4 the authors write "In their observational study, Kidwell et al. (2016) used Open Data Badges to reward researchers for sharing their data which increased the data sharing rate at the journal Psychological Science from 1.5% to 39% (13)." From this study, it is not possible to conclude that the badges actually increased data sharing rate. This is merely a correlation, so a better way of phrasing this would be "In their observational study, Kidwell et al. (2016) used Open Data Badges to reward researchers for sharing their data which was related to an increased data sharing rate at the journal Psychological Science (from 1.5% to 39%) (13)."

RESPONSE: Changed the wording of this sentence on page 4: In their observational study, Kidwell et al. (2016) used Open Data Badges to reward researchers for sharing their data which

was related to an increased data sharing rate at the journal *Psychological Science* (from 1.5% to 39%).

Similar cases in which causal conclusions are implied where they are not justified are:

* “A study published in 2017 found that while badges rapidly increased data sharing between 2013 and 2015 at *Psychological Science*, ...”

RESPONSE: Changed the wording of this sentence slightly on page 4: A study published in 2017 found that while badges were associated with a rapid increase in data sharing between 2013 and 2015 at *Psychological Science*...

* An observational study that examined the effect of badges at the journal *Biostatistics*, found that badges increased data sharing by only 7.6% and had no effect on code sharing (15).

RESPONSE: Changed the wording of this sentence slightly on page 4: An observational study that examined the effect of badges at the journal *Biostatistics*, found that badges were associated with increased data sharing of only 7.6% and had no effect on code sharing (15).

* “So, badges worked, but had a limited efficacy.”

RESPONSE: Changed the wording of this sentence slightly on page 4: So, badges were related to a slight increase in data sharing, but had a limited efficacy.

* “These findings contrast to those of Kidwell et al. (2016) who found open data badges increased data sharing from 1.5% to 39% at *Psychological Science* (13).”

RESPONSE: Changed the wording of this sentence slightly on page 16: These findings stand in contrast to those of Kidwell et al. (2016) who found that open data badges were associated with an increase in data sharing from 1.5% to 39% at *Psychological Science* (13).

* “A possible explanation behind these striking differences in badge effects [...]”

RESPONSE: Changed the wording of this slightly: A possible explanation behind these striking differences in data sharing rates....

* “Previous literature has shown that open data badges were effective in motivating researchers to share their data”

RESPONSE: Changed the wording of this sentence slightly on page 19: Previous literature has shown that open data badges were associated with motivating researchers to share their data however the findings of our trial demonstrate otherwise.

A related case is the last sentence: “Badges might still have the potential to change data sharing behaviour but they must be introduced with signalling or as a part of an encouragement ‘bundle’.” I may have missed it, but this suggestion for an intervention also seems to be based on observational data and speculation.

RESPONSE: Changed the wording of this sentence slightly on page 19: Badges might still have the potential to change data sharing behaviour but the way in which they are introduced, for example with signalling or as a part of an encouragement ‘bundle’, could be associated with such a change.

Power

It is unfortunate that the authors were not able to obtain the statistical power they were aiming at. Their estimated power of 60% is very low, and it is important to mention this limitation clearly

in the limitations sections. It is possible that the low power is an explanation for the lack of effect in the study (although the authors should be careful with such post-hoc explanations of the results, of course).

RESPONSE: We have emphasized the width of the confidence interval in the results and the discussion and this illustrates that our study did not have the power to rule out a potentially worthwhile badge effect.

The 60% power was calculated based on an effect size that was half of what has been detected in previous studies. What would be the power of this study to detect the same effect as in previous studies? I think it would be interesting to mention this at the end of the paragraph about power.

RESPONSE: We calculated the power for detecting the same effect as the previous badge study and added a sentence to the end of the paragraph on page 5: However, if we wanted to detect the same effect size as the previous badges study, that is a 37.5% effect, 80 papers per group gives us a 99.7% power to detect the 8% versus 37.5% difference in data sharing.

Operationalization “data sharing”

The authors explicitly mention that the data had to be shared on a third-party site. What if the data were shared as supplemental materials in BMJ Open itself? Would that count as a third-party site as well? Or doesn't BMJ Open allow such supplemental materials anyway?

RESPONSE: The BMJ Open recommends that authors share their data on Dryad. It is not usual that researchers would deposit data as supplementary material as those files are typically PDFs or images that cannot be re-analyzed. Data are typically shared at a third-party site as it will be easily downloadable and accessible by all (publicly available).

As a second point, the authors required that the data had to be “accessible”. This needs some further explanation. What does “accessible” mean exactly? Was it also checked whether these were data that were actually relevant for the research question? Or would anything that looked like data be counted as open data, regardless of its relevance/completeness?

RESPONSE: Added in two sentences on page 7 that discuss accessibility in more detail: Accessibility was verified by downloading the dataset and ensuring that it could be opened and that it contained data. We did not check to ensure that these data were relevant or complete as we did not have access to the articles' abstract or full text to verify those factors

Similarly, on page 8, the authors also require the data to be “easily downloadable”. What does this mean?

RESPONSE: Added a sentence on page 8 that defines “easily downloadable”: (accessible), meaning that the data are able to be downloaded and opened up with no restrictions

Tables

There seem to be some reproducibility issues with the tables in the paper. In Table 1, the percentages of Type of Study, Article accepted for publication at BMJ Open, and Participants

withdrawn do not seem to add up to 100% in the different conditions. The same holds for Table 3, where the percentages in the control condition add up to 101%.

RESPONSE: These percentages were rounded to whole figures, that's why they didn't add up to exactly 100%. We have added a footnote to each table to explain this.

I also wonder if it might facilitate interpretation of the results if the data in Table 1 were (also) depicted in a figure. This may make it easier to see differences (or lack thereof) between the conditions.

RESPONSE: Added in a figure on page 12.

I think the interpretation of Table 3 is somewhat confusing. Are these categories that BMJ Open uses or did the authors create them based on reading all statements? Furthermore, I find it a bit confusing that Table 3 shows that 3 papers in the control condition had data available at a third-party repository, whereas the conclusion above was that only 2 papers received a badge. Does that mean that 1 paper had data in a third-party repository that were not easily downloadable?

in Table 3, I think it would be clearer to include all 4 data sharing categories in the table, not just 3. Now, there are 6

RESPONSE: The three data sharing categories were developed from our previous studies. From our previous studies we also understood further the nuances of data sharing where an author might have shared data but if the links were broken, then it was 'potentially available'. Data that are under controlled access also were categorized under 'potentially' available. We wish to keep the categories to three as they are straightforward and easy to understand: available, potentially available, not available.

As to your question which outlines why only 2 papers received a badge in the control group when 3 had shared data, this was clarified in our limitation section: "One researcher had their dataset under embargo until the article was in publication phase, which meant they could not receive a badge." Added in another sentence on page 12 to clarify that and to add that the researcher was in the control group.

Data statement analyses

As an additional analysis, the authors also looked at the differences in word type and number of words between data statements of the control and intervention condition. It is not entirely clear to me what such a quantitative analysis would tell us. Did the authors have any substantive reason to look at these characteristics of the statements? Similarly, I don't think Figure 2 adds a lot to the paper. I think the figure is not intuitive and I wonder what the message is that it should tell me.

Since I don't know why we would expect any difference in type of words or number of words of the data statement between the two conditions (and what difference specifically we would expect), I would like to suggest to remove these analyses out of the paper. If the authors feel that these analyses should stay, then I would like to see some justification in the paper on why this is interesting to look at, or what type of differences they expected.

RESPONSE: We have decided to keep these in the paper. Our expectation is that the intervention could have had an impact on the language people used in their data sharing statements. The fact that there was no change was important, and backs up the null finding from the primary outcome.

Data and code

I think it's really great that the authors shared their code in RMarkdown format. I could access and run the code with only two minor issues. First, in the R file BMJOpenBadges_R_2019-05-24_1042.r, there are some weird symbols on line 1, which causes an error if you try to source it. Furthermore, the code gives an error when I'm trying to generate the CONSORT diagram, because I didn't have a folder called "figures".

RESPONSE: We have added a warning to the github page about the need for subfolder called 'figures'. We could not see any odd characters at the top of our Rmarkdown file.

Minor remarks

* In the preregistration, the authors indicate that they plan to test if there is an interaction between intervention and study type on open data badges. I don't think I've seen this test in the paper. This is probably due to the (extremely) low number of papers that earned a badge, but I think it would be good if the authors would add a sentence to the manuscript stating why they didn't test the interaction anymore.

RESPONSE: Yes, this is correct. I have added in a sentence on page 13 explaining this: We had originally planned (in our pre-registered protocol) to test whether there was an interaction between the effect of badges (intervention or control arm) and study type, however, given the low number of papers that earned a badge, we did not examine this interaction.

* You could consider adding "RCT" to your key words. In my view, this is an important and unique characteristic of your study.

RESPONSE: Added this to the key words on page 1.

* p. 3 line 16: missing space: key component?

RESPONSE: Corrected this.

* p. 4: the authors state that data badges are the only incentive that has been used for data sharing, but what about journal requirements? Would that not also count as an incentive? For instance, the default policy of PLOS journals that data have to be shared? Or the data statement from BMJ itself? I also miss this option in the discussion where other incentives for data sharing are discussed. I would argue it could also be an interesting RCT to see if making data sharing mandatory has any effect?

RESPONSE: This is very interesting however I would argue that a policy change is not an incentive or a reward. It would be an interesting RCT to see if data sharing as a mandatory policy will have any effect but our paper is focusing on motivation and incentives.

* p. 4 line 12: I may be mistaken, but Kidwell and colleagues did not actually hand out the badges, did they? The phrase "Kidwell et al. used Open Data Badges to reward research" implies that they did.

RESPONSE: Yes, they did give out badges (electronic badges on papers).

* p. 4 line 54: "A sample of 171 papers per group would have given us a 90% power to detect a

difference in data sharing of 8% (based on five published studies)”, I’m not sure what the authors mean with “based on five published studies”?

RESPONSE: These five published studies were previous studies that looked at baseline data sharing rates at journals. Rephrased the sentence slightly on page 5: A sample of 171 papers per group would have given us a 90% power to detect a difference in data sharing of 8% (average baseline data sharing rate at journals based on five published studies)

* p. 4 line 56: I’m surprised the authors calculated power based on a two-sided test. They had a clear one-sided hypothesis, so I’m wondering if a one-sided test wouldn’t be justified here.

RESPONSE: This is a debatable point, however we went with the norm of a two-sided test. In such a complex area there is always the possibility of unintended consequences. For example, imagine if our email with the promise of a badge spurred the authors to have a group discussion with other investigators about data sharing. Given the often negative reactions to data sharing, such a discussion could have led to reduced data sharing.

* p. 5 line 44: “A few papers that did not meet the exclusion criteria were wrongly included ...” How many?

RESPONSE: This sentence should read “A few papers that *did meet* the exclusion criteria were wrongly included...” This number was small – we no longer have it unfortunately.

* p. 7 line 37: could the authors provide a (ballpark) inter-rater reliability here? And I’m also curious to hear what caused the discrepancies.

RESPONSE: The very low rate of data sharing meant that almost all of the papers were very easy to judge as “no sharing”, hence there was very little to disagree about.

* p. 8 line 48-50: the authors mention something about contacting the authors of the studies they included. What did they have to contact them for? I don’t see this mentioned anywhere else in the paper.

RESPONSE: This is correct, we had originally anticipated that we would have to contact authors about their datasets but we no longer had to do that in order to verify shared data. I have removed the bit of the sentence that says that we collected data on time needed to contact authors. We did however collect data on amount of time needed to verify data.

* p. 9 line 7: “202 papers consented ...”, I would rephrase this. Papers cannot consent to anything ;)

RESPONSE: Rephrased this to read “...authors of papers consented...”

* p. 11 line 7-9: missing word in “Papers in the control group who shared their data at a public repository were also offered [an Open Data Badge?] after the trial ended.”

RESPONSE: Corrected this, yes it was an Open Data Badge.

* p. 15 line 23: typo: participants

RESPONSE: Corrected this.

* p. 16: “One such example of a funding body incentivizing data sharing is Berlin Institute of Health (BIH) and BIH QUEST Center rewarding scientists at Charité - Universitätsmedizin Berlin and the Max Delbrück Center for Molecular Medicine for disclosing the original data of their publications with a total of 200,000 euros (22).” I think I’m misunderstanding this sentence. It seems to say that if you share your data, you get 200,000 euros. That surely can’t be true? Maybe sharing the data is a requirement for getting a grant of 200,000 euros? Perhaps rephrase.

RESPONSE: It is a bit confusing, however the article does make it seem that researchers who share their data will receive 200,000 euros. <https://www.bihealth.org/en/notices/bih-rewards-open-data-in-an-effort-to-make-science-more-verifiable/>. Although, we think it means 200,000 Euros to share amongst all those who share. Changed the wording slightly: One such example of

a funding body incentivizing data sharing is Berlin Institute of Health (BIH) and BIH QUEST Center rewarding scientists at Charité - Universitätsmedizin Berlin and the Max Delbrück Center for Molecular Medicine for disclosing the original data of their publications with a total sum of 200,000 euros to be divided amongst all those who share (22)

* p. 16: “the trial results could have overestimated the effect of badges”. This seems impossible given that the observed effect was zero.

RESPONSE: Yes, but this is a limitation of our study, it had the potential of over-estimating the effect of badges due to the opt-in method.

Appendix B

Dear Andrew, Denes and Michèle,

Thank you for review and for provisionally accepting the paper for publication at Royal Society Open Science. Please find our responses to Editor and referees below in blue.

Kind regards,
Anisa, Adrian A and Adrian B

25-Feb-2020

Dear Dr Rowhani-Farid:

On behalf of the Editors, I am pleased to inform you that your Manuscript RSOS-191818.R1 entitled "Did awarding badges increase data sharing in BMJ Open? A randomized controlled trial" has been accepted for publication in Royal Society Open Science subject to minor revision in accordance with the referee suggestions. Please find the referees' comments at the end of this email.

The reviewers and Subject Editor have recommended publication, but also suggest some minor revisions to your manuscript. Therefore, I invite you to respond to the comments and revise your manuscript.

- Ethics statement

- Data accessibility

If you wish to submit your supporting data or code to Dryad (<http://datadryad.org/>), or modify your current submission to dryad, please use the following link:
<http://datadryad.org/submit?journalID=RSOS&manu=RSOS-191818.R1>

- Competing interests

- Authors' contributions

- Acknowledgements

- Funding statement

Because the schedule for publication is very tight, it is a condition of publication that you submit the revised version of your manuscript before 05-Mar-2020. Please note that the revision deadline will expire at 00.00am on this date. If you do not think you will be able to meet this date please let me know immediately.

When submitting your revised manuscript, you will be able to respond to the comments made by the referees and upload a file "Response to Referees" in "Section 6 - File Upload". You can use this to document any changes you make to the original

manuscript. In order to expedite the processing of the revised manuscript, please be as specific as possible in your response to the referees.

on behalf of Dr Denes Szucs (Associate Editor) and Essi Viding (Subject Editor)
openscience@royalsociety.org

Associate Editor Comments to Author (Dr Denes Szucs):
Associate Editor: 1

Comments to the Author:

Please revise all points raised by the reviewers.

Reviewer comments to Author:

Reviewer: 1

Comments to the Author(s)

Dear Authors,

Thank you very much for addressing my remarks and your efforts to revise the manuscript. I still have some issues that should be addressed before publication:

Major remarks:

1. I am still not convinced by the hypothesis concerning the secondary outcome; why should this be the case from a theoretical perspective? There should be at least some arguments why the used terminology in the data availability statement is expected to differ.

RESPONSE: The hypothesis behind these secondary outcomes is that the intervention could have influenced researchers to be more conscious of data sharing and a measure of that is what they write in their final data sharing statements which can be studied by looking at the differences between the types of words used and the number of words. The fact that there were no differences between the types of words and the number of words supports the null finding for the primary outcome.

On page 9 under Secondary outcomes, we have added this: "A potential secondary effect of badges is changes in what authors write in their data sharing statements. For example, the intervention might have prompted authors to be more detailed in explaining why they could or could not share their data."

2. In the abstract, please report the OR in the same reporting style as in the paper.

RESPONSE: On page 2, paragraph 2 (results section of the abstract), I added these sentences:

"...the odds of awarding badges were nearly equal in the intervention and control groups (Odds Ratio = 0.9, 95% CI [0.1, 9.0]). Data sharing rates were low in both groups, with just two data sets shared in each of the intervention and control groups."

Minor remarks:

1. The numbering of the paragraphs all starts with 1, is this intentional?

RESPONSE: I did not add these numbers in - I believe this issue was caused by ScholarOne.

2. p.17 l.18 consider the term reanalysis instead of "can repeat analyses to"

RESPONSE: We prefer our original wording.

3. p.23 l.36 change "the only number eligible for a badge is number 4" to "papers are only eligible if they provide direct access to the data (category 4).

RESPONSE: Re-adjusted the wording on page 9, paragraph 1: "Thus, papers were only eligible for a badge if they provided direct access to data at a public repository, the 'Directly available' group (category 4)."

4. The paper may need some format editing as some captions are numbered and some are not (e.g. "other outcome" on p.24 l.20). Also, the terminology should be checked as "other outcomes" are also referred to as "secondary outcomes" (p.29 l.14).

RESPONSE: On page 9, we replaced “Other outcomes” with “Secondary outcomes” and we added caption numbering for Secondary outcomes (1.6.2). We also added another section called “Other data” (1.6.3).

Reviewer: 2

Comments to the Author(s)

I would like to thank the authors for their responses to both my comments and the ones of the other reviewer. I only have a couple of minor points, but other than that, I am looking forward to see this paper in print.

Signed,
Michèle Nuijten

I first would like to say that I agree with the authors’ response to reviewer 1 that they decided to keep the methodological details in the paper. These details are important to have in order to be able to interpret the findings.

Next, I would like to clarify my point about Table 3 and the different categorizations of open data. I appreciate the authors’ wish to keep the categories they developed in previous studies. I think I mixed up the categories of data sharing with the categories of data sharing *statements*. That said, the reason I brought it up in the first place, is that several different sets of classifications are used throughout the paper that are not all used in presenting the results.

First, the authors introduce four different categories:

A categorical data sharing outcome was made using the categories:

1. No data deposited at a public repository
2. Data deposited at a public repository but data could not be verified due to embargoes and/or broken links
3. Author(s) stated that data are available only after further applications (e.g. ethics), or other conditions (e.g. payment, travelling to host institution), or upon reasonable request. Or author(s) stated that data are available via a repository where access is provided only after a proposal to use data has been approved by an access committee, and within the terms of a data sharing agreement. Hence, data are available under controlled access.
4. Data deposited in public repository without restriction

Next, they summarize these four categories in three other categories:

Based on the above we also tabulated the data as

1. ‘Not available’ (categories 1, 2),
2. ‘Potentially available’ (category 3) and
3. ‘Directly available’ (category 4).

(As an aside, if I understand the text in the manuscript correctly, the authors would consider a broken link “not available”, not as “potentially available” as they state in their response.)

These three categories are then further classified in:

1. Open Data Badge (only category 3)
2. No Open Data Badge (all other categories)

Unless I overlooked it, there are only results available for the final, dichotomous categorization: whether papers did or did not earn a Data Badge. I think that instead of presenting different sets of classifications, it might be clearer if the authors would present their dichotomization first, and then explain which situations would be eligible for a badge and which wouldn't. Otherwise, the reader might still be expecting the results for the four or three different categories that were presented earlier in the paper. It is of course possible that only I experienced this confusion, in which case incorporating this suggestion is not necessary.

RESPONSE: Sorry for the confusion, but we think that before we present the dichotomization which describes who would be eligible for a badge and who wouldn't, prior knowledge of how we made that decision is important: a presentation of categorical data sharing outcomes, and then classification based on previous studies and then finally dichotomous categorization. Accordingly, we have decided to maintain the current way we have presented our categories. We note that this complexity arises from the many different ways people share their data, but the need for the badge to be based on a straight dichotomy.

Another point I made in my first review, was that it was unclear to me why the authors did a word count analysis in the data statements. In their response (and in the paper), the authors explain they were interested in seeing whether there was a difference in the language used between conditions. I see that reviewer 1 was also not convinced by the relevance of these analysis (see point 7 in his review), which signals to me that the rationale of this analysis is not sufficiently explained. If the authors are adamant on keeping this section in the paper, I would suggest adding some additional explanation in which the authors give some examples of the possible changes in language they expected to find (e.g., perhaps the authors expected that the intervention caused people to be more detailed in explaining why they couldn't share the data). Such examples may help convince the reader that these secondary analyses can give interesting insights in potential side-effects of giving out badges.

RESPONSE: We would like to keep the planned secondary outcome exploring differences in number of words between control and intervention. The hypothesis behind these secondary outcomes is that the intervention could have influenced researchers to be more conscious of data sharing and a measure of that is what they write in their final data sharing statements which can be studied by looking at the differences between the types of words used and the number of words as well. The fact that there were no differences between the types of words and the number of words supports the null finding for the primary outcome.

On page 9 under Secondary outcomes, we have added this: “A potential secondary effect of badges is changes in what authors write in their data sharing statements. For example, the intervention might have prompted authors to be more detailed in explaining why they could or could not share their data.”

Then, the weird symbols in the R script were not in the Markdown file, but in the R file in the data folder (BMJOpenBadges_R_2019-05-24_1042.r). On line 1, it reads: `ï»¿#Clear existing data and graphics.`

RESPONSE: We cannot see the symbols in this R script file, to us it looks like: “#Clear existing data and graphics” and we have checked this online via Github and in R.

Finally, some additional comments about the minor remarks:

* p. 4 line 12: I may be mistaken, but Kidwell and colleagues did not actually hand out the badges, did they? The phrase “Kidwell et al. used Open Data Badges to reward research” implies that they did.

RESPONSE: Yes, they did give out badges (electronic badges on papers).

Reply to the authors: What I meant was that the paper implies that Kidwell and colleagues personally handed out badges, whereas the actual badges were awarded by the journal *Psychological Science*. Kidwell et al. just recorded whether a paper did or did not receive a badge.

RESPONSE: We have clarified that Kidwell et al. worked in collaboration with the journal to reward authors with badges. We have adjusted the text to read: “In their observational study, Kidwell et al. (2016), in collaboration with the journal *Psychological Science*, used Open Data Badges to reward researchers for sharing their data which was related to an increased data sharing rate at *Psychological Science* (from 1.5% to 39%) (13).”